

# REMIND2.1: Transformation and innovation dynamics of the energy-economic system within climate and sustainability limits

Lavinia Baumstark[1], Nico Bauer[1], Falk Benke[1], Christoph Bertram[1], Stephen Bi[1], Chen Chris Gong[1], Jan Philipp Dietrich[1], Alois Dirnaichner[1], Anastasis Giannousakis[1], Jérôme Hilaire[1], David Klein[1], Johannes Koch[1], Marian Leimbach[1], Antoine Levesque[1], Silvia Madeddu[1], Aman Malik[1], Anne Merfort[1], Leon Merfort[1], Adrian Odenweller[1], Michaja Pehl[1], Robert C. Pietzcker[1], Franziska Piontek[1], Sebastian Rauner[1], Renato Rodrigues[1], Marianna Rottoli[1], Felix Schreyer[1], Anselm Schultes[1], Bjoern Soergel[1], Dominika Soergel[1], Jessica Strefler[1], Falko Ueckerdt[1], Elmar Kriegler[1], Gunnar Luderer[1]

1Potsdam Institute for Climate Impact Research (PIK), Member of the Leibniz Association, P.O. Box 60 12 03, 14412 Potsdam, Germany

*Correspondence to*: Lavinia Baumstark (baumstark@pik-potsdam.de)





**Abstract.** This paper presents the new and now open-source version 2.1 of the REgional Model of INvestments and Development (REMIND). REMIND, as an Integrated Assessment Model (IAM), provides an integrated view on the global
energy-economy-emissions system and explores self-consistent transformation pathways. It describes a broad range of possible futures and their relation to technical and socio-economic developments as well as policy choices. REMIND is a multi-regional model incorporating the economy and a detailed representation of the energy sector implemented in the General Algebraic Modeling System (GAMS). It uses non-linear optimization to derive welfare-optimal regional transformation pathways of the energy-economic system subject to climate and sustainability constraints for the time horizon 2005 to 2100. The resulting solution corresponds to the decentral market outcome under the assumptions of perfect foresight of agents and internalization of external effects. REMIND enables analyses of technology options and policy approaches for climate change mitigation with particular strength in representing the scale-up of new technologies, including renewables and their integration in power markets. The REMIND code is organized into modules that gather code relevant for specific topics. Interaction between different modules is made explicit via clearly defined sets of input/output variables. Each module can be represented by different realizations enabling flexible configuration and extension. The spatial resolution of REMIND is flexible and depends on the resolution of the input data. The framework can thus be used for a variety of applications in a customized form balancing requirements for detail and overall run-time and complexity.

# 1 Introduction

This paper presents the new and now Open Source version 2.1 of the REgional Model of INvestments and Development (REMIND). The focus is predominantly on the technical structure and the representation of processes in REMIND. Further, illustrative results are presented. The Integrated Assessment Model (IAM) REMIND was originally introduced by (Leimbach et al., 2010b). This paper is an update of previous documentation of the model version 1.5 (Luderer et al., 2013), version 1.6 (Luderer et al., 2015), and version 1.7 (IAMC).

The first chapter provides an overview of REMIND as an Integrated Assessment Model. In chapter 2 the regional and temporal resolution of REMIND, its modular code structure, interfaces with other models and the solution algorithm are presented. The representation of different sectors and processes are described in chapter 3. Chapter 4 shows some exemplary results, while chapter 5 discusses the strengths and limitations of REMIND.

## 1.1 What are IAMs?

Integrated Assessment Models (IAMs) provide an integrated view of the global energy-economy-climate-land system. By asking questions like "can the world still reach the 2 degree target, under which socio-economic conditions and applying which technological options?", it is the goal of these models to explore self-consistent transformation pathways of these highly interdependent subsystems. IAMs can spell out a broad range of possible futures and their relation to technical and socio-economic developments as well as policy choices. More specifically, IAMs are mostly used for sustainable



transformation and development pathway analysis and exploring climate policy and technology options. Some IAMs are
based on intertemporal optimization as a powerful and valuable methodological approach since it enables the derivation of
optimal policies to be used as benchmarks in the analyses of other policy options. These analyses constitute an important part
of international reports on climate change including the works from the IPCC (Rogelj et al., 2018b) and the UNEP gap
reports (UNEP, 2019).

Shared by many IAMs, the Shared Socio-economic Pathways (SSPs) and Representative Concentration Pathways (RCPs)
provide a common reference framework for assumed socio-economic developments and greenhouse gas emission levels
(O'Neill et al., 2013). The use of SSPs helps to cover uncertainties regarding technological development for renewable or
fossil fuel availability, but also social and behavioral development like population growth, dietary preferences,
environmental awareness or international cooperation.

The history of integrated assessment modeling dates back several decades (van Beek et al., 2020) and by now a wide range
of different integrated assessment models are available. They differ in their level of detail, structure, solution method, and
time horizon, and are continuously being developed, which makes categorization difficult (Krey, 2014). Nevertheless, some
IAMs are derived from top-down macroeconomic models such that a stylized energy system is embedded into a
macroeconomic modeling framework, while other IAMs stand in the tradition of systems engineering models and take a
bottom-up perspective on the energy system, which comes at the cost of macroeconomic detail. Hybrid IAMs (Hourcade et
al., 2006) aim at combining a solid macroeconomic framework with high process detail of mitigation options. The latter is
required to describe systems transformations that take into account path dependencies and explicit technological
development. By contrast, there are some top-down models that are dedicated to cost-benefit analyses of climate mitigation,
requiring an even broader modelling scope including climate damages, which comes at the cost of any explicit representation
of process-based mitigation options (e.g. DICE (Nordhaus, 2010), FUND (Anthoff and Tol, 2013)). Whereas process-based
IAMs typically take a cost-effectiveness approach, in which a given climate target is reached at minimal economic costs of
climate mitigation, the REMIND model represents some damages and can thus be used for cost-benefit analyses or least total
cost analyses (as presented in (Schultes et al., 2020a)) (see section 3.1.3).

### 1.2 What is REMIND?

REMIND is a modular multi-regional model with a detailed representation of the energy sector in the context of long-term
macro-economic developments (see fig. 1). REMIND enables the exploration of a wide range of plausible developments and
of possible futures of the energy-economic system exploring self-consistent transformation pathways. REMIND can be
coupled to the land use model MAgPIE (see section 2.4.1) and the climate model MAGICC (see section 2.4.3) for a full
integrated assessment of the energy-economy-land-climate system. In this paper the version REMIND 2.1.3 is presented and
used for the production of outputs.
REMIND is implemented as a nonlinear programming (NLP) mathematical optimization problem. Its algebraic formulation
is implemented in GAMS (GAMS, 2020). CONOPT version 3.17 (CONOPT, 2020) is used as the numerically efficient





solver for the NLP problem. R (R Core Team, 2019) is used for code management as well as handling of input data and postprocessing. REMIND calculates economic and energy investments for an intertemporal Pareto optimum in the model regions for the time horizon 2005 to 2100, fully accounting for interregional trade in goods, energy carriers and emissions

allowances. REMIND enables analyses of technology options and policy proposals for climate change mitigation, along with sustainability challenges related to development, air pollution and - via coupling to MAgPIE (Dietrich et al., 2019; Dietrich et al., 2020) - land-use.

The macro-economic core of REMIND (Leimbach et al., 2010b; Leimbach et al., 2010a; Bauer et al., 2012b; Luderer et al., 2012) features a multi-regional general equilibrium representation of the Ramsey growth model, i.e. the investment share of

economic output is determined endogenously to maximize intertemporal welfare. This approach is well-suited for describing patterns of long-term economic growth (e.g., convergence between developing and industrialized countries) (Barro and Sala-i-Martin, 2004), which are key drivers of energy demand and thus emissions. The optimization is subject to equilibrium constraints, such as energy balances, economic production functions or the budget constraint of the representative household. The model explicitly represents trade in final composite good, primary energy carriers, and if certain climate policies are

enabled, emissions allowances. Equilibrium thereby refers to the balance in goods markets and international trade, such as the global oil market. It is a valid assumption for the decadal timescales considered in scenarios, and thus does not compromise the validity of the model dynamics. REMIND is usually run in a decentralized mode where each model region is optimized separately, and clearing of global trade markets ensured via iterative solutions (see section 2.2).

The macro-economic production factors are capital, labour, and final or useful energy. A nested production function with

constant elasticity of substitution determines the energy demand. REMIND uses economic output for investments in the macro-economic capital stock as well as for consumption, trade, and energy system expenditures. The macro-economic core and the energy system part are hard-linked via final or useful energy demand (input to the economy) and the costs incurred by the energy system (output of the economic part). Economic activity results in demand for energy in different sectors (transport, industry, buildings) and of different types (electric and non-electric). The primary energy carriers in REMIND

include both exhaustible and renewable resources. Exhaustible resources comprise coal, oil, gas and uranium. Renewable resources include hydro, wind, solar, geothermal, and biomass. More than 50 technologies are available for the conversion of primary energy into secondary energy carriers as well as for the distribution of secondary energy carriers into final energy.

REMIND uses reduced-form emulators derived from the detailed land-use and agricultural model MAgPIE (Lotze-Campen et al., 2008; Dietrich et al., 2019) to represent land-use and agricultural emissions as well as bioenergy supply and other

land-based mitigation options. REMIND can also be run in soft-coupled mode with the MAgPIE model (see section 2.4.1).

The model accounts for the full range of anthropogenic greenhouse gas (GHG) emissions, most of which are represented by source. REMIND simulates emissions from long-lived GHGs ($CO_2$, $CH_4$, $N_2O$), short-lived GHGs (CO, NOx, VOC) and aerosols ($SO_2$, BC, OC). It calculates $CO_2$ emissions from fuel combustion and industrial processes, $CH_4$ emissions from fossil fuel extraction and residential energy use, and $N_2O$ emissions from energy supply based on sources. F-Gases and

emissions from land-use change are included exogenously with different trajectories depending on SSP and climate target.



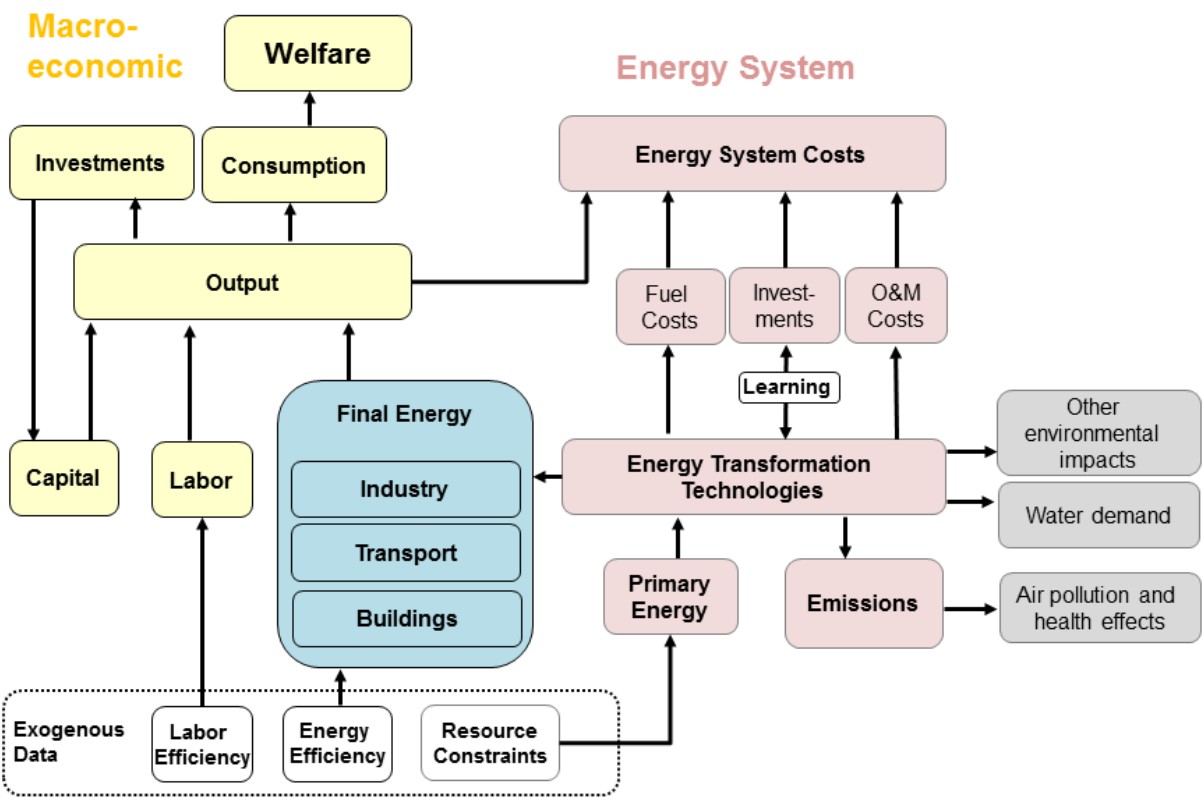

**Figure 1: Structure of REMIND**

In terms of its macroeconomic formulation, REMIND resembles other well-established integrated assessment models such

as RICE (Nordhaus and Yang, 1996) and MERGE (Manne et al., 1995). However, REMIND is broader in scope and features

a substantially higher level of detail in the representation of energy-system technologies, trade, and global capital markets.

Its comparative advantage of the high technology detail allows a more detailed exploration of efficient strategies to attain an

exogenously prescribed climate target ("cost-effectiveness mode").

Scenarios developed with previous REMIND versions were published in numerous studies (Bauer et al., 2012a; Bertram et

al., 2015; Strefler et al., 2018a). REMIND was also part of various model inter-comparison projects (e.g. ADVANCE

(Luderer et al., 2018), CD-LINKS (Roelfsema et al., 2020), EMF-30 (Harmsen et al., 2019), EMF-33 (Bauer et al., 2018,

p.33), SSP (Riahi et al., 2017)) as well as the international research initiative for developing the SSPs. The scenario data are



accessible via the databases hosted at IIASA (e.g. the IAMC 1.5°C Scenario Explorer (Huppmann et al., 2018)). The scenarios and SSP framework were used for international assessment processes (IPCC, 2018; IPCC, 2019; The World in

2050 initiative (TWI), 2018). Some of these studies included dedicated diagnostic exercises to assess the dynamic behaviour of the models (Kriegler et al., 2015), or focused on comparing input assumptions across models (Krey et al., 2019).

### 1.3 Inputs and outputs of REMIND

REMIND uses a range of exogenous data as an input to ensure consistency of scenarios with historic developments and realistic future projections. Historical data for the year 2005 is used to calibrate most of the free variables (e.g. primary

energy mixes in 2005, secondary energy mixes in 2005, standing capacities in 2005, trade in all traded goods for 2005). Technology parameters are projected into the future, in general assuming a certain level of convergence across regions in the very long term. Projections of possible future demographic and economic developments offer population and labour trajectories from 2005 to 2100 (SSP trajectories (Dellink et al., 2017; KC and Lutz, 2017)). To align with GDP trajectories consistent with the population trajectories from 2005 to 2100 (see fig. 2) and with final and useful energy trajectories,

REMIND calibrates its production function as described in section 2.3.

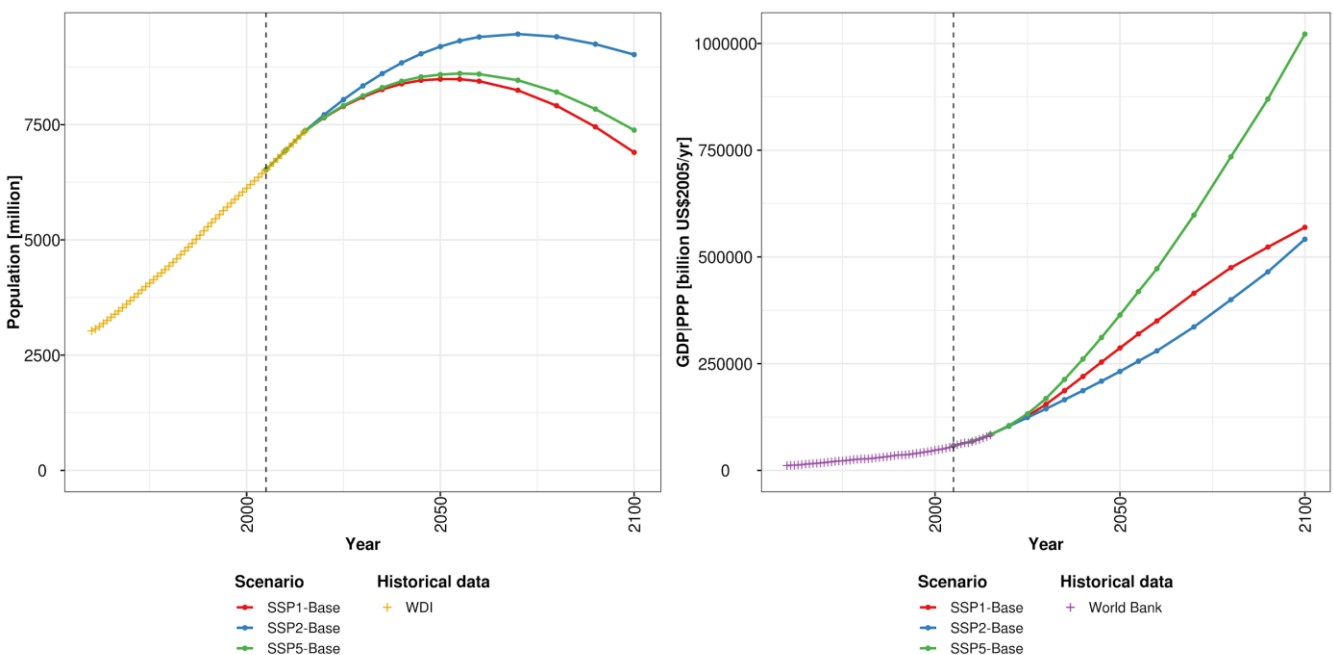

**Figure 2: Global population and GDP trajectories for 2005 to 2100 for SSP1, SSP2 and SSP5, compared to historical data from the World Development Indicator (WDI) and (World Bank, 2012).**

Based on these input parameters REMIND calculates investments into different technological capacities and capital until

2100, price-induced adjustments of final energy use, the resulting primary and secondary energy trajectories, emissions of all





greenhouse gases and imports and exports of traded goods until 2100. This enables the analysis of technology options and policy proposals for climate change mitigation.

## 2 Code structure and general modelling philosophy

The REMIND code is structured in a modular way, with code belonging either to the model core, or to one of REMIND's
modules. A module gathers all code relevant for a certain topic and interacts with other modules or the core through a clearly defined set of input/output variables only (interface). Each module can be represented by different realizations. This structure allows for more complex or simple realizations of each module as long as all interfaces (i.e. incoming and outgoing information) between the modules and the core are addressed in a suitable way. Depending on the research questions to be analysed, a different realization of a module can be used. For example, if the focus is on the fossil fuel sector a realization
with detailed representation of this sector would be chosen. In most other applications a realization with an emulator of the complex version with less computational demand is used (for more information about the modular structure see Dietrich et al., 2019 - Appendix A.

REMIND is run by executing scripts in R, which take the file "main.gms", load configuration information and build the model, by concatenating all necessary files from the core and modules folders into a single file called "full.gms".
This paper focuses on realizations which are active in default scenarios. More detail about all modules and their interlinkages can be found in the model documentation (https://rse.pik-potsdam.de/doc/remind/2.1.3/) (Luderer et al., 2020b).

### 2.1 Spatial and temporal discretization and input data management

REMIND is an intertemporal optimization model, deriving an equilibrium solution of the world economy under the assumption of perfect foresight. The spacing of time steps is flexible. In the default case, there are five-year time steps until
2060, ten-year time steps until 2110 and twenty-year time steps after that. The analysis of scenarios is typically focused on the time span 2005-2100, but the model runs until 2150 to avoid distortions due to end effects.

Also the spatial resolution of REMIND is flexible. It depends on the resolution of the input data, which is computed separately from the GAMS code. Using the R-packages "mrremind " (Baumstark et al., 2020), "mrcommons" (Bodirsky et al., 2020) and "madrat" (Dietrich et al., 2017) it is possible to generate the input data for any spatial aggregation of ISO-
country specific data.

By default REMIND calculates results for the 12 following world regions: CAZ -Canada, Australia, New Zealand; CHA - China; EUR - Europe; IND - India; JPN - Japan; LAM - Latin America; MEA - Middle East and Africa; NEU - Non-EU Europe; OAS - Other Asia; REF - Reforming Economies; SSA - Sub-Saharan Africa ; USA - United States of America. Countries from the same territorial area and/or similar development level and/or similar climate policies are merged into the
same world region. Some countries which are of specific interest regarding climate change mitigation (e.g. USA, CHN, IND) are represented individually.



For research projects focussing on specific areas/regions (e.g. Europe, Australia) REMIND can be run with higher spatial resolution (i.e. more than the 12 global default regions). By parallelizing the calculation of the individual regions in decentralized optimization mode (see section 2.2) the computation time increases only moderately with increasing spatial
detail.

In practice, there are some limitations to the spatial resolution. First, it is not guaranteed that the model will find an optimal solution for a new region. Second, for each new spatial resolution the plausibility of the results needs to be checked (especially for very small countries), as some country-specific peculiarities might not be fully captured by the general model structure.

**2.2 Solution algorithm**

REMIND is formulated as an intertemporal optimization problem. Time represents a separate dimension. The solution algorithm in the module "optimization" optimizes over all time periods simultaneously, hence deals with time in the same way as with other dimensions. In essence, the time dimension only increases the number of markets for which the algorithm has to find an equilibrium. As part of the overall optimization problem, REMIND is searching for a distinguished
equilibrium related to the trade interaction between countries and regions. Based on economic concepts (Walrasian tatonnement process, Negishi method (Negishi, 1972)), two algorithms (Nash and Negishi) are developed and used to find a competitive equilibrium and PARETO equilibrium, respectively (Leimbach et al., 2017). Manne and Rutherford (1994) applied the Negishi approach in an intertemporal setting using a joint maximization algorithm (which is similar to the present algorithm). For the numerical process REMIND is using the CONOPT solver, which is supposed to find a local optimal
solution. It is not sure that a global optimum is reached, but the stability of the equilibrium by running the model with different initial values is monitored. In the course of thousands of experiments nearly exclusively unique solutions are observed.

In Nash mode [realization "nash" of the module "80_optimization"], each region forms its own optimization problem. Regions trade in goods and resource markets, but market-clearing conditions are not part of the optimization itself. Instead,
regions are subject to an intertemporal budget constraint. The Nash-algorithm iteratively computes solutions for all regions including their trade patterns, and adjusts prices such that the surplus on global markets vanishes. Initial values for trade patterns, prices etc. are taken from former solutions.

Benefits of a Nash-solution are a massive reduction in run-time (thanks to the possibility of parallel computing, scenarios converge within one to a few hours), and more flexibility in treating inter-regional externalities. Learning-by-doing
technologies are included by default and cause an inter-regional spill-over. In Nash-mode, a subsidy on the investment cost of learning technologies can be used to internalize this spill-over externality [realization "globallyOptimal" of module "22_subsidiseLearning"] (Schultes et al., 2018). Without internalizing the learning-by-doing spill-over due to the global technology learning curves, Nash and Negishi solutions differ.





In Negishi mode [realization "negishi" of the module "80_optimization"], all regions form a single optimization problem
(global welfare maximization with iteratively adjusted regional welfare weights). Regions trade in goods and resource
markets, and market-clearing conditions are part of the optimization. The Negishi algorithm computes solutions
simultaneously for all regions including regional trade patterns, and between iterations adjusts the so-called Negishi weights
until a Pareto optimal solution without transfers is found. Lending and borrowing across regions is allowed, but
intertemporal trade balances need to be equalized. Regional utilities are summed up weighted by the Negishi weights to form
the global welfare function of REMIND.

## 2.3 Calibration of the production function

REMIND uses a nested production function with constant elasticity of substitution (CES) to determine a region's gross
domestic product (GDP). The module "29_CES_parameters" covers two options: the calibration of parameters of the
production function [realization "calibrate"] and the loading of former parameters for this function [realization "load"].
Inputs at the upper layer of the production function include labour, capital, and energy services. Labour is represented by the
population at working age. Energy services at the upper level are the output from a CES tree combining sectoral energy
inputs from transportation, buildings and industry. In turn, the demand for specific energy carriers at the sectoral level is also
depicted through individual CES nests. Each production factor in the various macroeconomic CES functions has an
efficiency parameter. The aim of the CES calibration [realization "calibrate" of module "29_CES_parameters"] is to provide
the efficiency parameters of the CES tree for each time step and each region. The changes of efficiency parameters over time
are tuned such that the baseline scenario meets exogenous economic growth pathways (Dellink et al., 2017) and final or
useful energy pathways (see section 2.4.2) in line with the SSPs (O'Neill et al., 2014).

The calibration has to fulfil two constraints: an economic and a technological one. The technological constraint requires the
inputs of the CES function to yield the desired output. At this stage, there is no economic consideration at all. During a
REMIND run however, the model will strive to find the most efficient solution in terms of costs. Therefore, the second
constraint is an economic constraint. The derivatives of the CES function, i.e. the marginal increase in income from
increasing the considered input by one unit, must equal the price of that input, i.e. the marginal cost.

The calibration operates in several iterations. In each iteration the nested CES function is adapted such that the exogenous
final energy pathways and the exogenous GDP and labour trajectories are matched. Each iteration only differs from the
others in the prices that are provided to the calibration, which are the feedback from the energy system. The efficiency
parameters converge towards a stable set of values.

The economic constraint defines that the prices are equal to the derivatives. The technological constraint determines,
following the Euler's rule, that, for homogeneous functions of degree one (as it is the case here), the output is equal to the
sum of the derivatives times the quantity of inputs. Combining both constraints means that the output is equal to the sum of
inputs valued at their price. So, the prices and quantities given exogenously, combined with the two constraints, are
sufficient to determine all the quantities of the CES tree up to the last level with labour and capital.





For many assumptions on variables which influence the macro-economic dynamic of REMIND (e.g. SSP-scenario) CES parameters already exist and can be loaded [realization "load" of the module "29_CES_parameters"].

### 2.4 Interfaces with other models

The model REMIND can be coupled to other models that have more detail in specific areas (see fig. 3). The coupling interfaces are usually soft links and lead to a consistent solution by running the respective models after each other and updating some information iteratively. The "Energy Demand GEnerator" (EDGE) (Levesque et al., 2018) models inform REMIND about final energy demands while "Model for the Assessment of Greenhouse Gas Induced Climate Change" (MAGICC)(Meinshausen et al., 2011) calculates radiative forcing and global mean temperature based on emissions from

REMIND. The interface with MAgPIE (Lotze-Campen et al., 2008; Dietrich et al., 2019) enables the analysis of consistent land use scenarios.

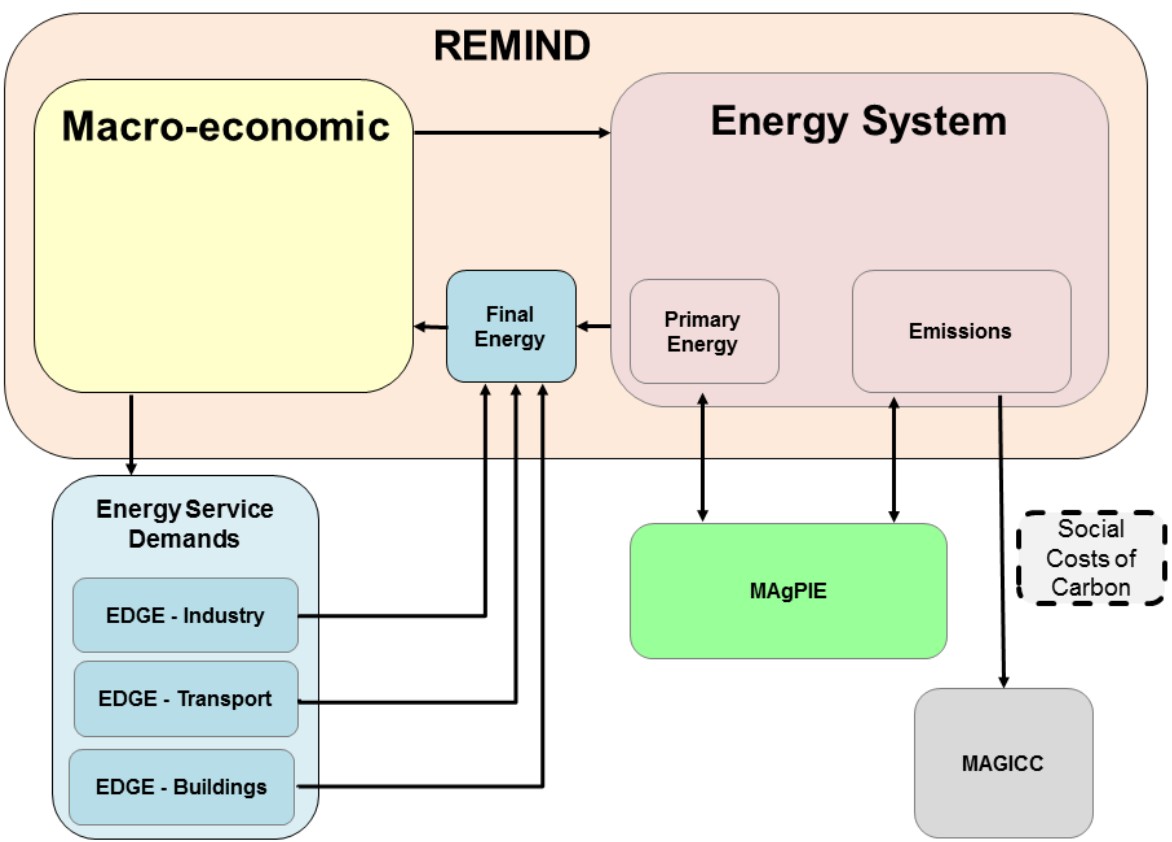

**Figure 3: REMIND and possible links to other models**





### 2.4.1 Land use (MAgPIE)

From a climate protection perspective, two aspects of the land-use sector are of particular interest: the supply of biomass that can be used for energy production (possibly with carbon capture and storage - CCS) and the total emissions of the land-use sector. By default REMIND uses supply curves for purpose-grown biomass, and exogenous projections for land use emissions and agricultural production costs as described in section 3.2.5. These projections have been derived from the land-use model MAgPIE (Lotze-Campen et al., 2008; Dietrich et al., 2019) for a set of the most common climate targets (Representative Concentration Pathways - RCPs) and socio-economic development pathways (SSPs). Only for these scenarios the assumptions on the land-use quantities in REMIND are consistent with MAgPIE. When changing crucial parameters in REMIND (such as the climate target or availability of technologies or resources) this can have significant impact on GHG prices and bioenergy demand, such that the assumptions on the land-use parameters mentioned above would not be consistent anymore with the response of the land-use system. Thus, to cover these potential deviations from the standard scenarios REMIND can be run in an iterative soft-coupled mode with MAgPIE (Klein, 2015; Bauer et al., 2020), where REMIND updates MAgPIE's assumptions on bioenergy demand and GHG prices and MAgPIE in turn updates REMIND's assumptions on bioenergy prices and land-use emissions and agricultural production costs. The iteration is continued until changes between iterations become negligible. The resulting scenarios are consistent regarding price and quantity of bioenergy and GHG emissions.

### 2.4.2 Deriving baseline energy demand pathways from sectoral EDGE models

Energy demand pathways depend on numerous drivers and constraints which vary across energy sectors (transportation, industry, buildings), but also across sub-sectors. The determinants of the demand for space heating and cooking differ as much as do the determinants for steel and chemical production. To limit the complexity of the model, REMIND does not represent all variables and parameters that would be relevant for the future development of energy demand. Instead, detailed sectoral EDGE models (EDGE-Buildings, EDGE-Transport, EDGE-Industry) produce final energy pathways. The baseline scenario of REMIND, which assumes no climate policy, is calibrated to meet these final energy pathways (see section 2.3). In policy scenarios, the demand would then evolve in reaction to the effects of carbon prices and other price shifts.

    a) EDGE-Transport

        Beside the default realization "complex" of the module "35_transport" (see section 3.3.1), REMIND can run coupled to the transport model EDGE-Transport (Rottoli et al., accepted).

        To represent transport sector demands, EDGE-Transport has been engineered, as a successor of the "Global Change Analysis Model" (GCAM) transport module (Mishra et al., 2013; Kyle and Kim, 2011), to interface with REMIND. The detail required to model fine-grained sector specific dynamics would add too much of a burden to the REMIND optimization routine.



The coupling with EDGE-Transport significantly increases the level of detail on the technological and modal choice. It also adds further criteria to the decision-making process. Actual consumer decisions are governed by both tangible costs as well as other decision drivers. The mobility consumer in EDGE-Transport is susceptible to time invested in traveling (Schafer and Victor, 2000), range anxiety (Bonges and Lusk, 2016), inertia of the infrastructure system (Waisman et al., 2013), consumer lifestyles (Le Gallic et al., 2017), and the availability of models.

The consistency between REMIND and EDGE-Transport is achieved via two distinct steps. First, the baseline demand for transport energy services in REMIND's production function is calibrated to the baseline projections from EDGE-Transport for all regions and time steps. Second, REMIND and EDGE-Transport are solved iteratively to ensure consistency between the prices and quantities of energy services required by the transport system. In the iterative process, EDGE-Transport informs REMIND about the market shares gained by the different transportation technologies, as well as the per-unit costs and per-unit energy intensity of each node. On the basis of this information, REMIND determines the volume of energy services demand for transport.

On the REMIND side of the coupling, transportation demands are represented as strongly aggregated categories: transport is divided into passenger and freight demand, which each include a short-to-medium and a long-distance option. The aggregated demands are accounted for in energy service units (ton kilometer for freight, passenger kilometer for passenger transport), as the benefit to households and firms results from the amount of travelling and transported goods. EDGE-Transport provides the initial configuration of demand for each production factor for the model calibration phase, where the set of efficiency parameters is calculated for the baseline economic and technological development scenario (see section 2.3).

b) EDGE-Industry

Final energy demand for the industry sector is based on trajectories tuned to conform to experts' judgement of future developments in the sector in absence of climate change mitigation policies. The original eleven-region time series are disaggregated to country level, adjusted to follow recent historic trends for a period until mid of the century, and again aggregated to the desired regional resolution. REMIND is then calibrated to meet these trajectories in the baseline scenario (see section 2.3).

c) EDGE-Buildings

The future of buildings energy demand will depend on manifold factors including demographic and socio-economic trajectories, but also climate, floorspace demand, and buildings components. Because of the diversity of relevant factors and the limited resources to include them all in the REMIND model, for computational reasons, buildings energy demand projections are split into a two-step process. First, the EDGE-Buildings model (Levesque et al., 2018; Levesque et al., 2019)— a detailed buildings bottom-up model — is used to project energy demand in the absence of climate policy. The REMIND baseline scenario is calibrated (see section 2.3) to this trajectory. Second, in the climate policy scenario, building energy demand in REMIND reacts to carbon pricing by adjusting the energy





315 demand level as well as the distribution among energy carriers, with a typically higher demand for electricity in climate scenarios. The EDGE-Buildings model is therefore only run before calibrating the REMIND model, and not between REMIND run iterations as is the case for the EDGE-Transport model.

### 2.4.3 Climate (MAGICC)

REMIND calculates GHG emissions from different sectors such as energy production, transport, land use change and waste.
320 To translate emissions into changes in atmospheric composition, radiative forcing and temperature increase, REMIND can be coupled with the MAGICC 6 (Meinshausen et al., 2011) climate model [realization "magicc" of module "15_climate"]. Due to numerical complexity, the evaluation of climate change using MAGICC is performed after running REMIND. Iterative adjustment of emission constraints or carbon taxes allows meeting specific temperature or radiative forcing limits in case of temperature targets.

325 **2.5 Exploring scenarios - most common climate policy scenarios**

REMIND is able to explore a wide range of plausible developments of the energy-economic system using the concept of perfect foresight. The model provides an integrated view of possible futures of the global energy-economy system exploring self-consistent transformation pathways. The focus of these scenarios is on climate change mitigation in the cross-sectoral context under consideration of technological and socio-economic changes. But those self-consistent scenarios are not to be
330 understood as forecasts, but projections that depend on a broad set of assumptions, including policies (Nakicenovic et al., 2000). Applying perfect foresight is a powerful methodological approach to derive first-best, benchmark scenarios for reaching climate targets. Those benchmark scenarios enable the analysis and comparison of different policy scenarios and serve as the basis of policy advice. Real-world investment decisions - by energy corporations for instance - are guided by expectation formation, which is typically based on intertemporally-optimizing planning tools.

335 An alternative to the perfect foresight assumption is that of adaptive expectation formation. This approach hypothesizes that economic agents always assume that prices remain constant and base their investment decisions on this simple extrapolation. As prices change earlier, investments turn out regrettable and adjustments are made in the next period. It is well-known that the adaptive expectation assumption leads to cyclical investment behaviour, huge swings in prices and unstable technology deployment patterns. On the contrary, the perfect foresight assumption implies a rational expectation equilibrium that leads
340 to stable long-term development.

The perfect foresight assumption of REMIND holds for various parts, not only intertemporally, but also across regions and sectors (i.e. emission reductions happen first where they are cheapest). But at least as important as the provision of perfect benchmark scenarios is the ability of REMIND to limit foresight and generate scenarios featuring imperfections. In this case, REMIND operates in a mode of false expectations (e.g., regarding the stringency of climate policies) to analyse pathways
345 that are intertemporally sub-optimal. In a number of REMIND studies such settings have been applied, e.g. in the context of delayed action scenarios (Jakob et al., 2012) or limited technological availability (Luderer et al., 2013). Moreover, the





effects, if international spillovers are not fully internalized in technology support policies are implemented and discussed in (Schultes et al., 2018). Similarly, recent developments of REMIND account for short-sightedness of certain agents, e.g. the owner-renter relationship in the buildings-sector (Levesque et al., 2021) or consumer choice in transportation (Rottoli et al.,

accepted). Those policy scenarios do not have complete perfect foresight, but only some element of foresight under scenario constraints.

With different bundles of such scenarios, the model can address various research questions. For each scenario, the model calculates cost-optimal investments in economy and energy sectors by maximizing global welfare subject to equilibrium constraints. By default, negative impacts of climate change are ignored (see section 3.1.3 for options for representing

damages), but the representation of the full basket of GHGs allows calculating the temperature outcome of each scenario.

Baseline scenarios without any explicit representation of climate policies serve as benchmarks and for the purpose of final energy calibration. In addition, regularly computed climate policy scenarios include scenarios following current country plans (nationally determined contributions - NDCs), National Policies implemented (NPi), and stylized policy scenarios with different ad-hoc assumptions on policy stringency and burden-sharing, each described in more detail below.

A scenario which follows the NDCs as submitted to the UNFCCC between 2015 and 2017 is implemented by a stylized representation of technology policies and targets for a few major regions and countries, and emission constraints based on quantifiable country targets, achieved via iteratively adjusted regional carbon prices. Both the technology targets and the emissions targets are implemented in a separate module [realizations "NDC2018" of the modules "40_techpol" and "45_carbonprice"]. Most targets are implemented for the year 2030, and a middle-of-the-road assumption is taken for

extrapolation of policy stringency beyond that year: sectoral targets are moderately strengthened, and carbon prices are assumed to moderately increase and gradually converge until 2100.



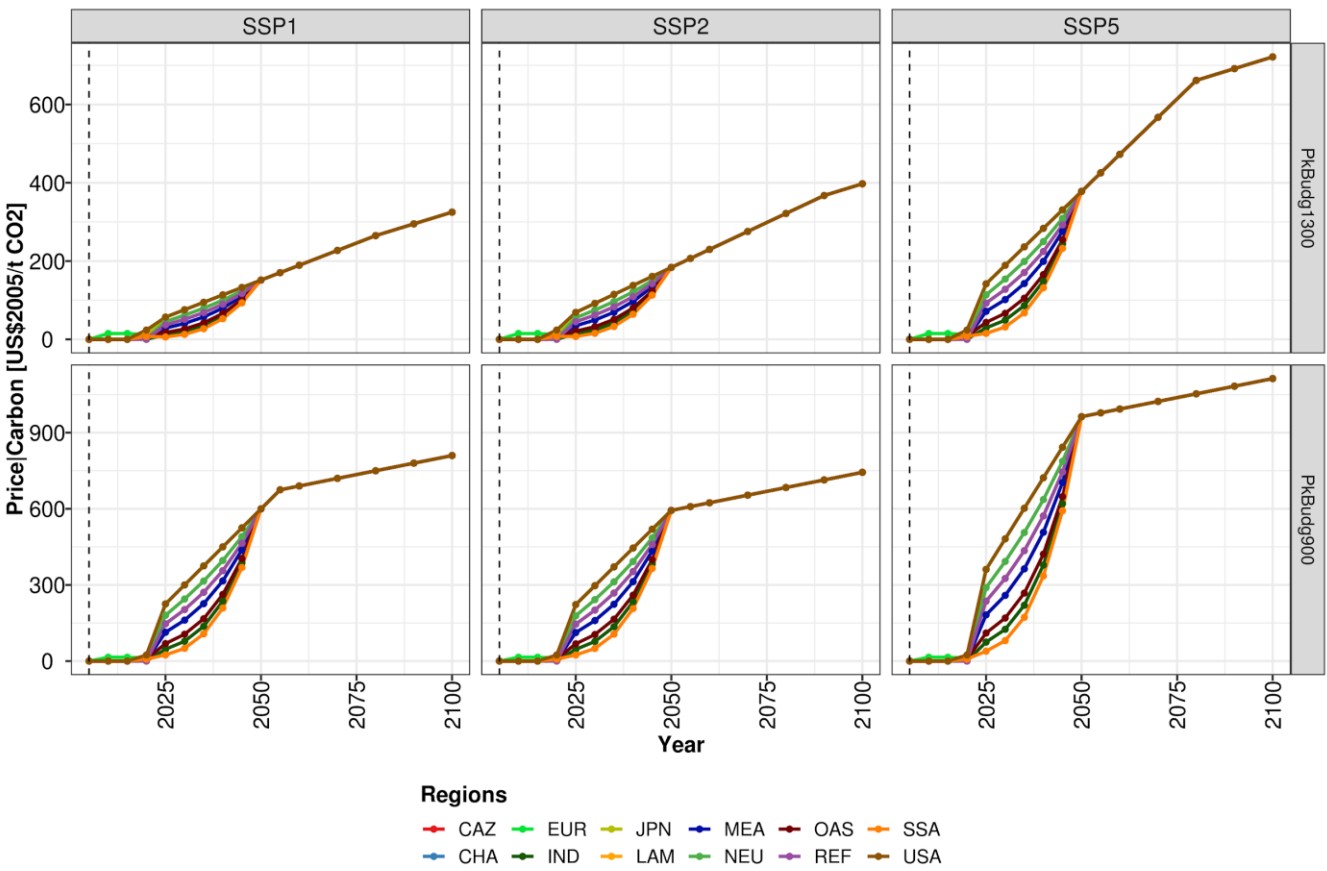

**Figure 4: regional CO2 price trajectories for NPi and PkBudg900 scenarios for SSP1, SSP2 and SSP5**

The current policy scenario (NPi) is identical to the NDC scenario until 2020 (via fixing of variables), but assumes policies
fail to achieve NDC targets in 2030. Instead, carbon prices are assumed to grow and converge more slowly, leading to
emissions trajectories in line with bottom-up studies on the effect of currently implemented policies (den Elzen et al., 2019).
Stylized climate policy scenarios either assume an explicit carbon price trajectory, or a bound on cumulative emissions, i.e. a
budget on total $CO_2$ emissions from 2011-2100, in Gt CO2. The most commonly used budgets rely on the IPCC 1.5°C
report, chapter 2, Table 2.2 (Rogelj et al., 2018b). By introducing a carbon price (see section 3.1.2 on implementation of
taxation) which is iteratively (after each Nash or Negishi iteration) adjusted the carbon budget is met. The carbon price
transitions to the level consistent with the long-term policy starts after 2020. The carbon price adjustment can be
instantaneous (jumping to the new value within one time step), or with a period of gradual convergence implemented as
regional differentiated carbon prices (see for example (Kriegler et al., 2018)). In the latter scheme, developing countries
initially face much lower prices, but gradually converge to a globally uniform price level. As a default setting for REMIND
2.1, a carbon price differentiation according to GDP/cap [PPP] values in 2015, and a convergence of regional carbon prices
until 2050 is used (see fig. 4).





In 1.5°C scenarios, peak warming is allowed to be at or slightly above 1.5°C, at median climate sensitivity (MAGICC 6 (Meinshausen et al., 2011)), but returns to values below 1.5°C with at least 67% by the end of the century (Rogelj et al., 2018a). With default SSP2 settings, this is implemented via a peak-budget of 900 Gt CO2 from 2011 until time of net-zero CO2 emissions, with slightly net-negative emissions thereafter so that end-of century budgets are around 700-800 Gt. For well below 2°C scenarios, the peak budget is typically set to 1300 Gt CO2. The peak budget approach (Rogelj et al., 2019) is represented in REMIND by a specific shape of the carbon price trajectory, with a steep linear increase in the front-runner regions (see above for the default regional carbon price differentiation and convergence) until the peak budget is reached, and a further slow linear increase of carbon prices at 3$ per year thereafter. The timing of the peak year, as well as the required carbon price in this year, are endogenously determined based on the peak-budget value. Thereby scenarios with high overshoot of the carbon budget around mid-century and large reliance on carbon dioxide removal (CDR) in the second half of the century, as they are common when $CO_2$ budgets are only specified for the year 2100, are avoided.

To account for uncertainty in input data (parametric uncertainty) REMIND is used in sensitivity analyses of techno-economic inputs (Bauer et al., 2018; Giannousakis et al., 2020b). REMIND is also used to run myopic scenarios (Luderer et al., 2013), as well as in a stochastic version (Giannousakis et al., 2020a) to account for uncertainty in the representation of the energy-economy-emissions system and socioeconomic/regulatory uncertainty about the future.

## 3 System representation

In this section the representation of different processes which are implemented in REMIND are described. Most of the different aspects of the model are separated into modules of REMIND and can be described by different realizations.

### 3.1 Macro-economy

#### 3.1.1 Drivers of economic growth

The macro-economic core of REMIND features a multi-regional general equilibrium growth model (Barro and Sala-i-Martin, 2004). This model is well suited to describe patterns of long-term economic growth (e.g., convergence between developing and industrialized countries), which are key drivers of energy demands and thus emissions. Physical capital is a major driver of economic growth and related investments are endogenous in such models. The representative agent, endowed with perfect foresight, has in each period to make the choice of using output for consumption or for investment, which is consumption tomorrow. Perfect foresight is a standard assumption in economic models and widely used IAMs (e.g. DICE/RICE (Nordhaus and Yang, 1996), MERGE (Manne et al., 1995), MESSAGE (Fricko, 2016), WITCH (Bosetti et al., 2007)). While in real world agents rarely have perfect foresight, using this concept is a useful approximation in a context of models with long planning horizons (see also discussion in section 2.5). While using the perfect foresight assumption to formulate an intertemporal optimization problem, the model is completed by components (technically - side constraints), that help to reproduce real world dynamics caused by imperfectly foresighted decision-making (for example adjustment costs for





the increase of the macroeconomic capital stock). In REMIND each region maximizes its welfare subjected to a budget constraint. The relevant equations are spread between the modules "02_welfare" and "01_macro". The "02_welfare" module implements an utilitarian social welfare function. Social welfare is equal to the discounted intertemporal sum of utility, which itself is a function of per capita consumption. Air pollution generated by the energy system induces a welfare penalty. The time preference rate, a parameter describing how much consumption in the future is valued compared to consumption in the present, and the intertemporal elasticity of substitution, a measure of the willingness to consume in the present instead of in the future, determine the trade-off between consumption today and in the future. While the discount rate equals the assumed time preference rate, the real rate of interest emerges endogenously according to the Keynes-Ramsey rule based on the two preference parameters and the optimal consumption growth rate.

### 3.1.2 Steady state and equilibrium

In economics, the long-term economic growth is called "steady state" and it is common to differentiate between steady state and equilibrium. While the steady state is a long-term property and related to the stability of the evolution problem (note: in contrast to physical sciences, "steady state" in the context of macro-economic growth theory means that key characteristics of the system, such as the savings rate, income share of labor, etc., remain constant, while the overall economy still grows), the equilibrium is a short-term concept. If an economic system is stable, a deviation from the steady state growth path leads to transition processes that close the gap to the steady state asymptotically. During this process the markets are in equilibrium (i.e. prices equal demand and supply) in each time step. This ensures that basic accounting requests are met (i.e. no loss of commodities at the global level). It may help to give an example. Consider an economy that is on a long-term steady state growth path based on fossil fuels. If this system is interrupted by a policy to reduce fossil fuel use while energy demand remains high, final energy prices rise, which is the expected shift in the short-term market equilibrium. These higher prices trigger investments into alternatives such as renewable energy. It takes some time to increase the capacity to produce non-fossil energy, but over time the final energy prices decrease as non-fossil energy supply ramps-up and the economy reverts back to the steady state. Hence, the equilibrium outside of the steady state makes those investments that move the economy closer to the steady state competitive. This steady state needs not to be the original steady state, because the supply of non-fossil fuels may have changed long-term economic growth for better or worse, but the economy approaches the long-term steady state, and during this transition the energy markets are in short-term equilibrium. The REMIND model is supposed to analyse such transition dynamics in response to policies.

It is possible to compute the Pareto-optimal global equilibrium including inter-regional trade as the global social optimum using the Negishi method (Negishi, 1972), or the decentralized market solution among regions using the Nash concept (Leimbach et al., 2017) [module "80_optimization"].

REMIND follows an equilibrium concept that is based on General equilibrium theory and Walras' law (Arrow and Debreu, 1954; Debreu, 1970; Ewing et al., 2006). By introducing intertemporal budget constraints for each country and world region, Walras' law is met, i.e. the value of excess demand is always zero. The general equilibrium, i.e. equilibrium prices that





equalize supply and demand in all markets, is then achieved by an iterative price adjustment process (Walrasian tatonnement process). The equilibrium is achieved instantaneously. Yet, based on the aggregated level of REMIND, this equilibrium just represents a balance of demand and supply of aggregated goods over time spans of 5 years.

The general equilibrium concept on which REMIND is based is mathematically and numerically tractable and the fundamental theoretical framework of a majority of economic models. It aggregates a large number of separate decisions by individuals in a way that coordinates production and consumption activities, balances supply and demand, and leads to an efficient allocation of goods and services in the economy - an outcome that to a large degree also characterizes real-world interactions. Yet, this concept also has some limitations. On the one hand, there are strong assumptions like the perfect information for all agents. On the other hand, uniqueness and robustness of the equilibrium cannot be demonstrated for a

very general set of assumptions (Balasko, 2009). The ability of REMIND to model long-term growth dynamics and ensuing energy demands is hardly contained by limitations of the equilibrium concept. Application of this concept is contained to international trade interactions, while the dynamics of long-term growth is mainly driven by preferences, productivities, technological change, capital accumulation, population growth and endowments (e.g. fossil resources).

Arrow and Debreu (1954) introduced with the general equilibrium theory also the two welfare theorems, according to which

a competitive market equilibrium can be determined as a Pareto optimum. This is exactly done with the Negishi approach that finds the equilibrium as a solution of a social planner problem.

### 3.1.3 Production and Trade

The module "01_macro" implements the macro-economic production, capital stock and GDP balance (or budget) equations. The production function is a nested CES (constant elasticity of substitution) function with capital, labour, and final energy

as inputs. Capital is enlarged by investments and depreciated according to the depreciation rate, labour is given exogenously, and energy is produced at a cost. Generated economic output (GDP) is used for consumption, investments in the macro-economic capital stock and energy system expenditures, as well as trade, non-energy related greenhouse gas abatement costs and agricultural costs delivered by the land use model MAgPIE (see sections 2.4.1 and 3.2.5). Tax revenues are redistributed as a lump sum, thus net taxes converge to zero in the optimal solution (equilibrium point).

REMIND considers the trade of coal, gas, oil, biomass, uranium, the composite good (aggregated output of the macro-economic system), and emissions permits (in the case of emissions-trading-system (ETS) based climate policy). It assumes that renewable energy sources (other than biomass) and secondary energy carriers are non-tradable across regions.

REMIND models regional trade via a common pool [module "24_trade"]. While each region is an open system - meaning that it can import more than it exports - the global system is closed. The combination of regional budget constraints and

balanced international trade (enforced by market clearing prices see section 2.2) ensures that the sum of regional consumption, investments, and energy-system expenditures cannot be greater than the global total output in each period. In line with the classical Heckscher-Ohlin and Ricardian models (Heckscher et al., 1991), trade between regions is induced by differences in factor endowments and technologies. REMIND also represents the additional possibility of intertemporal





trade. This can be interpreted as capital trade or borrowing and lending. Capital trade is linked to the export and import of
goods and energy, and is accounted for in the intertemporal trade balance. By directing the goods trade, the capital market
implementation affects the consumption.

To reconcile modelled capital flows and currently observed patterns (Lucas-Paradox - (Lucas, 1990)), REMIND represents
capital market imperfections [module "23_capitalMarket"]. The default setting includes limitations on the growth of debts
and surpluses each region can accumulate within a five-year period. As an alternative, a more comprehensive representation
of capital market imperfections is implemented. This realization considers imperfections on capital markets that in addition
to limits on debt accumulation take risk mark-ups on capital flows into account, which make lending of capital more costly
for some regions. Moreover, regionally differentiated preference parameters (so-called savings wedges) cover institutional
imperfections and help to further reconcile model results of short-term consumption and current accounts with observed data
(Leimbach and Bauer, 2020).

**3.1.4 Representation of taxes**

REMIND includes different types of taxes (see Table 1), representing existing energy taxes, emulating climate policies via
carbon prices or additional externalities for some technologies and processes. The representation of taxes is implemented in
the module "21_tax". The overall tax revenue is the sum of various components, each of which is calculated employing an
analogous structure: the tax revenue is the difference between the product of an activity level (a variable) and a tax rate (a
parameter), and the corresponding product from the last iteration (which is loaded as a parameter). After convergence of
Negishi/Nash iterations, the value of the tax revenue approaches 0, as the activity levels between the current and last
iteration do not change anymore. This means, taxes are budget-neutral: the amount of potential tax is always recycled back
and still available for the economy. Nevertheless, the marginal of the (variable of) taxed activities is impacted by the tax
which leads to the intended adjustment effect.


| tax type | rationale | calculation/implementation |
|---|---|---|
| bioenergy tax | represents negative externalities of bioenergy plantation on land | scales linearly with the bioenergy demand starting at 0 at 0EJ to the level defined in cm_bioenergy_tax at 200 EJ, tax rate (calculated as multiple of bioenergy price) times primary energy use of purpose-grown lignocellulosic biomass |
| greenhouse gas tax | main policy instrument for achieving mitigation targets | tax rate times GHG emissions |
| CCS tax | to represent performance difference of carbon stored in fuel vs. in form of $CO_2$ in geological storage | tax rate (defined as fraction (or multiplier) of operation and maintenance (O&M) costs) times amount of $CO_2$-sequestration |





| net-negative emissions tax | to represent marginal damages of overshoot in emissions budget (and temperatures) | tax rate (defined as fraction of carbon price) times net-negative emissions |
|---|---|---|
| final energy taxes in Transports | status quo of fuel taxation, with different assumptions on convergence | effective tax rate (tax - subsidy) times FE use in transport |
| final energy taxes in Buildings_Industry or Stationary | status quo of fuel taxation, with different assumptions on convergence | effective tax rate (tax - subsidy) times FE use in sector |
| final energy taxes in Buildings_Industry or Stationary sector with energy service representation | status quo of fuel taxation, with different assumptions on convergence | effective tax rate (tax - subsidy) times FE use in sector |
| resource extraction subsidies | status quo of extraction subsidies | subsidy rate times fuel extraction |
| primary to secondary energy technology taxes, specified by technology | represent not explicitly represented externalities of different technologies (water use, emissions of substances beyond $SO_2$ and $CO_2$) | effective tax rate (tax - subsidy) times SE output of technology |
| export taxes | represent export barriers | tax rate times export volume |
| $SO_2$ tax | represent air pollution externality | tax rate times emissions |
| high implicit discount rates in energy efficiency capital | mirror the overvaluation of initial investments vs. run-time costs by customers s | additional discount rate times input of capital at different levels |
| Regional subsidy on learning technologies | (only in nash runs): internalize the positive externality of the learning spill-over to other regions, so to arrive globally optimal solution, i.e. nash solution equivalent to negishi solution). | Subsidy for a technology is the sum over the regional capitalized benefits of learning which corresponds to the shadow price of the equation that describes the capacity build up of this technology. Conversion of this shadow price to a monetary value (dollar per watt) is achieved by normalizing with the shadow price of the budget equation. |

**Table 1: Tax types of REMIND and the reason why they are implemented and how**

### 3.1.5 Representation of economic damages due to climate change

Research on the economic impacts of climate change is rapidly evolving and there is no agreement yet on how exactly the effects of climate change affect the socioeconomic system. Traditional damage functions affect the level of output (e.g. the in DICE model (Nordhaus, 2017)). Empirical studies are now providing new top-down impact estimates with some evidence for possible effects of climate on growth rates (Burke et al., 2015). Applications show that the resulting compounding effects lead to much larger social costs of carbon and as a result more stringent mitigation action (Glanemann et al., 2020; Moore and Diaz, 2015). Reflecting this ongoing and open debate, REMIND uses a flexible approach to account for different types of macroeconomic damages.





Damages are included through a soft-coupled approach explained in detail in (Schultes et al., 2020a). Emissions from REMIND are passed on to the simple climate model MAGICC [realization "magicc" of the module "15_climate"] which calculates global mean temperature changes. These are passed to the damage module "50_damages" where different damage functions can be chosen to calculate the impacts. The reduction in output is passed back to the macro module "01_macro" and is included in the budget function as an exogenous parameter. In order to internalize the damage, the social cost of

carbon is calculated and included as a carbon price. Updating the social cost of carbon iteratively yields the same solution that a fully endogenous representation of climate and damages within REMIND would. The soft-coupled approach has two advantages. First, it allows more flexibility and complexity in the exogenous damage module. Second, it allows to easily combine damages with a climate target, reflecting that the available damage functions only include certain types of climate impacts (mostly productivity effects) and, in particular, omit tipping points and other potentially high impact processes to be

hedged against.

Currently, two different types of damages are implemented. The first are level effects, represented by four different specifications [realization "DiceLike"]: the function as used in the most recent versions of the DICE model (DICE2013R (Nordhaus, 2014) and DICE2016 (Nordhaus, 2017)), and two specifications from the meta-analysis of (Howard and Sterner, 2017).

The second type of damages are growth rate damages [realization "BurkeLike"]. One realization used the original empirical specifications by (Burke et al., 2015). The resulting GDP reduction of a one-off temperature shock is infinitely persistent in this formulation. In addition a specification introduced by (Schultes et al., 2020a) is included, where the GDP reduction has a finite persistence time only. This reflects the high uncertainty surrounding the empirical estimates and the possibility of future adaptation beyond historically observed degrees.

Regional temperatures are obtained through statistical downscaling based on CMIP5 (Taylor et al., 2012, p.5) results from the global mean temperature change pathway obtained from MAGICC. The temperature downscaling is based on the CMIP5 climate model ensemble and observed present-day temperatures calculated from the University of Delaware Air Temperature and Precipitation v4.01 data set (NOAA Physical Sciences Laboratory, 2020). Aggregation from gridded to regional temperatures uses constant 2010 population weights (Jones and O'Neill, 2016). Details are given in (Schultes et al., 2020a).

**3.2 Energy resources and supply**

**3.2.1 General representation of energy conversion and technologies**

The core part of REMIND includes the representation of the energy system via the conversion of primary energy into secondary energy carriers via specific energy conversion technologies. Around fifty different energy conversion technologies are included in REMIND. In general, technologies providing a certain secondary energy type compete linearly against each

other, i.e. technology choice follows cost optimization based on investment costs, fixed and variable operation and maintenance costs, fuel costs, emission costs, efficiencies, lifetimes, and learning rates. REMIND assumes full





substitutability between different technologies producing one energy type. Table 2 shows the secondary energy carriers included in REMIND and the sectors they are used in.

| | Industry | buildings | transport |
|---|---|---|---|
| Electricity | x | x | x |
| Hydrogen | x | x | x |
| Liquids | x | x | x |
| Solid fuels | x | x | |
| Gases | x | x | x |
| District heat and local renewable heat | x | x | |

**Table 2: Secondary energy carriers included in REMIND and the sectors they are used in**

A few technologies convert secondary energy into secondary energy, namely the conversion of electricity to hydrogen via electrolysis and the re-conversion via hydrogen turbines, as well as the production of methanol and methane from hydrogen.

In REMIND technologies are represented as linear transformation processes that convert one or more inputs into one or more outputs. In- and outputs can be energy, materials, water, intermediate products or emissions or labour inputs. The number of

in- and outputs is not restricted and technologies vary between in- and output characteristics. In the broader system context technologies and their deployment interact via various budget constraints, which give rise to competition for resources, but also the potential to expand feasible production possibilities. A model solution provides a set of activities that is feasible with all constraints simultaneously.

REMIND specifies each technology through a number of characteristic parameters

- Specific overnight investment costs that are constant for most technologies and decrease due to learning-by-doing for some relatively new technologies (see below).
- Cost markups due to financing costs over the construction time.
- Fixed yearly operating and maintenance costs in percent of investment costs.
- Variable operating costs (per unit of output, excluding fuel costs).

- Conversion efficiency from input to output.
- Capacity factor (maximum utilization time per year). This parameter also reflects maintenance periods and other technological limitations that prevent the continuous operation of the technology.
- Average technical lifetime of the conversion technology in years.
- If the technology experiences learning-by-doing: initial learn rate, initial cumulative capacity, as well as floor costs

- that can only be approached asymptotically.



REMIND represents all technologies as capacity stocks with full vintage tracking. Since there are no hard constraints on the rate of change in investments, the possibility of investing in different capital stocks provides high flexibility for technological evolution. However, the model includes cost mark-ups for the fast up-scaling of investments into individual technologies; therefore, a more realistic phasing in and out of technologies is achieved. The model allows for premature retirement of capacities before the end of their technological lifetime, and the lifetimes of capacities differ between various types of technologies. If capacities are phased out for economic reasons before they reach the end of their technical life-time, these assets are then stranded. Furthermore, capacities of conversion technologies age realistically from an engineering point of view: depreciation rates are very low in the first half of the lifetime and increase strongly thereafter.

In the modules "04_PE_FE_parameters" and "05_initialCap", each region is initialized with a vintage capital stock, and conversion efficiencies are calibrated to reflect the input-output relations provided by IEA energy statistics (Extended world energy balances (IEA, 2016)). The conversion efficiencies for new vintages converge across the regions from the 2005 values to a global constant value in 2050. Furthermore, for some fossil power plants, transformation efficiencies improve exogenously over time to represent technological advances. To match 2005 values in the IEA statistics, REMIND adjusts the regional by-production coefficients of combined heat and power (CHP) technologies.

### 3.2.2 Representation of exhaustible resources

REMIND characterizes the exhaustible resources coal, oil, gas, and uranium in terms of extraction cost curves [module "31_fossil"]. Fossil resources (e.g., oil, coal, and gas) are further defined by decline rates and adjustment costs (Bauer et al., 2016b). Extraction costs increase as low-cost deposits become exhausted (Herfindahl, 1967; Rogner, 1997; Aguilera et al., 2009; Bauer et al., 2016a). In REMIND, region-specific extraction cost curves that relate production cost increase to cumulative extraction (Bauer et al., 2016a; Rogner et al., 2012, p.7).

More details of the underlying data and method are presented in a separate paper (Bauer et al., 2016b). In the model, these fossil extraction cost input data are approximated by piecewise linear functions that are employed for fossil resource extraction curves. In the realization "timeDepGrades" it is possible to make oil and gas extraction cost curves time-dependent. This means that resources and costs may increase or decrease over time depending on expected future conditions such as technological and geopolitical changes. This representation is numerically and run-time demanding. Therefore, the default realization "grades2poly" of the module "31_fossil" emulates the supply generated by the time-dependent grades by polynomial functions. For uranium, extraction costs follow a third-order polynomial parameterization based on data of the Nuclear Energy Agency (NEA), see (Bauer et al., 2012a) for details.

### 3.2.3 Representation of renewable resources

REMIND models resource potentials for non-biomass renewables (hydro, solar, wind, and geothermal) using region-specific potentials in its "core". For each renewable energy type, potentials are classified by different grades, specified by capacity factors. Superior grades have higher capacity factors, which correspond to more full-load hours per year. This implies higher





energy production for a given installed capacity. Therefore, the grade structure represents optimal deployment of renewable energy, first using the best sites before turning to sites with worse conditions.

The renewable energy potentials of REMIND may appear higher than the potentials used in other models (Luderer et al., 2014). However, these models typically limit potentials to specific locations that are currently competitive or close to becoming competitive. The grade structure of REMIND allows for the inclusion of sites that are less attractive, but may become competitive in the long-term as the costs of technologies and fuels change. This choice is dependent on the model. The regionally aggregated potentials for solar photovoltaics (PV) and concentrated solar power (CSP) used in REMIND

were developed in (Pietzcker et al., 2014b) in cooperation with the German Aerospace Center DLR. To account for the competition between PV and CSP for the same sites with good irradiation, an additional constraint for the combined deployment of PV and CSP was introduced in REMIND (Pietzcker et al., 2014b) to ensure that the model cannot use the available area twice to install both PV and CSP.

The regionally aggregated wind potentials were developed based on a number of studies (Hoogwijk, 2004; Brückl, 2005;

Hoogwijk and Graus, 2008; EEA, 2009; Eurek et al., 2017). The technical potentials for combined on- and off-shore wind power amount to 800 EJ/year (half of this amount is at sites with more than 1900 full-load hours). The total value is roughly half as large as the maximum extractable electric energy from wind over land area as estimated in (Miller and Kleidon, 2016), and about one fifth of the potential estimated in (Lu et al., 2009).

The global potentials of hydropower amount to 50 EJ/year. These estimates are based on the technological potentials

provided in (WGBU, 2003). The regional disaggregation is based on information from a background paper produced for this report (Horlacher, 2003).

### 3.2.4 Representation of power sector and VRE integration

The realization "IntC" (IntC = Integrated Costs realization) assumes a single electricity market balance that is complemented with equations that implicitly represent challenges and options related to the temporal and spatial variability of wind and

solar power. The core approach (Pietzcker et al., 2014b) is an aggregated representation of technology and region-specific VRE integration costs and curtailment rates (i.e., unused surplus share of VRE electricity generation), which since 2017 are parameterized with the help of two detailed electricity production cost models (Scholz et al., 2017; Ueckerdt et al., 2017). Integration costs consist of costs associated with short-term storage deployment (batteries), long-term hydrogen storage (electrolysis and hydrogen turbines), transmission and distribution grid expansion and reinforcement, and curtailment of

surplus electricity. These drivers are parameterized for a range of wind and solar PV generation shares, as well as for the regional-specific temporal matching of electricity demand and renewable supply. In addition, operating reserve requirements are represented similar to a flexibility balance equation that was introduced for the MESSAGE model (Sullivan et al., 2013). In a more detailed representation "RLDC" (RLDC = Residual Load Duration Curve), the REMIND model represents regional load and renewable supply patterns in an explicit representation of RLDCs that endogenously change based on





regional VRE shares, exogenous battery and endogenous hydrogen storage, all of which is again parameterized with detailed electricity production cost models (Ueckerdt et al., 2017).

### 3.2.5 Representation of bioenergy - land use

The land-use sector is particularly relevant for climate change mitigation because of its big share of global emissions and its ability to provide the renewable and comparatively low-emission resource biomass. In REMIND, biomass is used to produce

the energy sources electricity, heat, ethanol, diesel, and hydrogen. Some of the conversion routes are equipped with CCS, which makes biomass an important source of negative emissions (Klein et al., 2014b). The following types of biomass are considered: food crops containing sugar, starch and oil; ligno-cellulosic residues from forestry and agriculture, and ligno-cellulosic grasses and trees from short-rotation plantations.

The latter is assumed to play a more important role in climate protection than biomass from food crops because of its

reduced adverse side effects on the land-use sector and the climate (food competition, deforestation, fertilizer, water consumption). Therefore, the resource potential for purpose grown lingo-cellulosic biomass is represented in REMIND via detailed supply curves (Klein et al., 2014a), while bioenergy from food crops is limited to today's level. The REMIND-MAgPIE coupling (see section 2.4.1) also focuses on ligno-cellulose from short rotation plantations.

The resource potential for the three biomass feedstocks is defined in the realization "30_biomass" of the module

"magpie_40". The price for purpose-grown ligno-cellulosic biomass is calculated as a (linear) function of demand according to the supply curves. The supply curves are exogenous to REMIND and have been derived in pre-processing by evaluating the price response of the MAgPIE model to different global bioenergy demand scenarios. Bioenergy costs of purpose-grown ligno-cellulosic biomass are calculated by integrating the price supply curve over the demand. Purpose-grown ligno-cellulosic biomass is the only biomass resource that can be traded between regions in REMIND. Residues from forestry and

food production are available as a limited low-cost lingo-cellulosic resource slightly increasing over time with a constant price.

Land use emissions are defined in the "core" as exogenous trajectories for $CO_2$, $CH_4$, and $N_2O$ derived from MAgPIE. They serve as emission baselines from which further abatement is possible according to the GHG price using marginal abatement cost curves (MACC). The MACCs for $CH_4$ and $N_2O$ are based on (Lucas et al., 2007) (see section 3.4.1 for details) .

Agricultural production costs (excluding costs of biomass production) are also exogenous scenarios for REMIND derived from MAgPIE and provided in the realization "costs" of the module "26_agCosts".

When coupled to MAgPIE the following measures are taken in REMIND to ensure consistency with the land-use system: the supply curves are updated by shifting them according to the price response of MAgPIE (Klein, 2015), the exogenous projections for land-use emissions, and non-biomass agricultural production costs are replaced with data from the latest

MAgPIE iteration. All land-use related MACCs are switched off in REMIND since abatement is realized in MAgPIE through changes in land-use patterns, technological change, and MACCs. Bioenergy trade remains in REMIND. Biomass from food crops is harmonized with MAgPIE in the pre-processing but is not part of the coupling.



### 3.3 Representation of energy demand sectors

#### 3.3.1 Transport

The module "35_transport" calculates the transport demand composition as a part of the CES structure. In the default realization "complex" transport demand composition is calculated for light duty vehicles (LDVs), electric trains and heavy duty vehicles (HDVs), an aggregate category including passenger non-LDVs and freight modes (Pietzcker et al., 2014a). The three corresponding nodes in the CES transport branch represent aggregated transportation demands in terms of useful, i.e., motive, energy. The LDV node in the CES tree is supplied by either electricity, hydrogen or liquid fuels with different

conversion efficiencies, accounting for vehicles with internal combustion engines, fuel cell cars or battery electric vehicles. The shares of the different drivetrain technologies are determined endogenously. HDVs can also be powered by liquid fuels, hydrogen and electricity; trains are all electric. REMIND keeps track of fleet capacities and accounts for additional costs per aggregated demand unit.

For a more detailed representation of the transport sector REMIND can be run coupled to EDGE-Transport (see section
2.4.2) by choosing the realization "edge_esm" of module "35_transport".

#### 3.3.2 Industry

The module "37_industry" models final energy use in the industry sector and its subsectors, as well as the emissions generated by them.

In the default realization "fixed_shares", the final energy demand is determined for the aggregated industry sector and
subdivided into four industry subsectors: cement production, chemicals production, iron and steel production, as well as all remaining industry energy demand (denoted 'other Industry') using region-specific shares that are kept constant at 2005 levels. Fuel switching (e.g. electrification) is enabled based on final energy prices and elasticities of substitution of the final energy carriers in the CES function.

In the realization "subsectors" the energy demand from industry is modelled explicitly for the four subsectors (cement,
chemicals, and iron and steel, as well as all remaining industry energy demand (denoted "other Industry") in the nested CES production function. The iron and steel sector is subdivided into primary steel (from iron ore) and secondary steel (from scrap). The production of cement and steel, as well as the value added from chemicals are derived via econometric regressions models based on per capita GDP at country level. Steel demand is projected following the approach of (Pauliuk et al., 2013).

In all realizations of the module "37_industry" three marginal abatement cost (MAC) curves have been derived from the literature for CCS in the cement, chemicals, and iron and steel sectors (Kuramochi et al., 2012). A fourth curve, that does not differentiate between the subsectors, was derived from (Fischedick et al., 2014). Subsector-specific MAC curves for CCS are applied to emissions calculated from energy use and emission factors according to the endogenous $CO_2$ price, to calculate industry $CO_2$ emissions and CCS. Process emissions from cement production are based on an econometric estimate of



cement production according to (Strefler, 2014) and are included in cement emissions for which CCS is applicable. Industry CCS costs (by subsector) are equal to the integral below the MAC cost curve.

### 3.3.2 Buildings

The module "36_buildings" determines the demand for final energy carriers necessary to provide energy services whose production will, in turn, determine the welfare of the representative consumer. In the default realization "simple", the

heterogeneity of the demand is rendered through a nested CES function with a high degree of substitutability among non-electric fuels (heating oil, natural gas, etc.) and a low degree of substitutability between non-electric fuels and electric demand. The distinction between the non-electric and electric energy carriers is motivated by the different uses that can be made of these energy sources. While non-electric fuels are mostly used for heating purposes (space, water and cooking), electricity consumption covers a wider range of purposes (lighting, appliances, cooling).

In addition to the default buildings representation, REMIND can also include the more detailed buildings realization "services_putty", that distinguishes not only between energy carriers but also across energy services with four categories ('appliances and lighting', 'space cooling', 'space heating', 'cooking and water heating'). Energy demand is not only depicted at the final energy level, but also at the useful energy level. The choice of energy carriers and technologies for heating purposes is dealt with outside the CES function to keep the physical balance between final and useful energy. The

choice is handled through a multinomial logit. The detailed module also includes a trade-off between efficiency investments and energy consumption for insulation, space cooling and appliances and can represent efficiency policies. Furthermore, the module includes a representation of the inertia dynamics at work in the buildings envelope investment cycle via a putty-clay formulation in the CES nested function (Levesque et al., in review).

The realization "services_with_capital" reproduces the features from the "services_putty" realization with the exception of

the specific inertia dynamics of the buildings envelope investments.

### 3.4 Representation of GHG emissions

REMIND simulates emissions from long-lived GHGs ($CO_2$, $CH_4$, $N_2O$), short-lived GHGs (CO, NOx, VOC) and aerosols ($SO_2$, BC, OC). REMIND accounts for these emissions with different levels of detail depending on the types and sources of emissions. It calculates $CO_2$ emissions from fuel combustion, $CH_4$ emissions from fossil fuel extraction and residential

energy use and $N_2O$ emissions from energy supply based on sources. The energy system provides information on the regional consumption of fossil fuels and biomass for each time step and technology. For each fuel, region and technology, REMIND applies specific emissions factors, which are calibrated to match base year GHG inventories (Global Emissions EDGAR v4.2, 2013; Amann, 2012).





### 3.4.1 Greenhouse gases

$CH_4$, $N_2O$, and $CO_2$ from land-use change have mitigation options that are independent of energy consumption and are calculated in the core of REMIND. However, there are costs associated with these emission reductions. Therefore, REMIND derives the mitigation options from marginal abatement cost curves (MACC), which describe the percentage of abated emissions as a function of the costs (Lucas et al., 2007). It is possible to obtain baseline emissions - to which the MACCs are applied - by three different methods: by source, by an econometric estimate, or exogenously. $CH_4$ fugitive emissions from

coal, oil, and gas extraction and processing, $CH_4$ from the residential sector, and $N_2O$ from energy supply are calculated by source using region- and fuel-specific emission factors. The emission factors for $CH_4$ fugitive emissions are derived using the emissions inventory (Global Emissions EDGAR v4.2, 2013) and the amount of fossil fuel extracted in each region in REMIND in 2005. Emission factors for $CH_4$ from the residential sector, and $N_2O$ from energy supply are taken from Amous (2000), Table 1. REMIND uses the econometric estimate for $CO_2$ emissions from cement production as well as $CH_4$ and $N_2O$

emissions from waste handling. In both cases, the driver of emissions depends on the development of the GDP (as a proxy for waste production) or capital investment (as a proxy for cement production in infrastructure). REMIND uses exogenous baselines for $N_2O$ emissions from transport and industry, and for $CO_2$, $CH_4$, and $N_2O$ emissions from land-use and land-use change based on MAgPIE (see section 3.2.5). $CH_4$ and $N_2O$ emissions from open burning are assumed to remain constant at their 2005 levels.

Emissions of other GHGs (e.g. F-gases, Montreal gases) are exogenous and are taken from the SSP scenario data set from the IMAGE model (van Vuuren et al., 2017). REMIND does not represent abatement options for these gases; therefore, emissions from the corresponding SSP/RCP scenario best matching the target of the specific model simulation are used.

### 3.4.2 Pollutants and non-GHG forcing agents

REMIND calculates emissions of aerosols and ozone precursors ($SO_2$, BC, OC, NOx, CO, VOC, $NH_3$) in the module

"11_aerosols". It accounts for these emissions with different levels of detail depending on sources and species.

For pollutant emissions of $SO_2$, BC, OC, NOx, CO, VOC and $NH_3$ related to the combustion of fossil fuels, REMIND considers time- and region-specific emissions factors coupled to model-endogenous activity data. BC and OC emissions in 2005 are calibrated to the GAINS model (Klimont et al., in prep.a; Amann et al., 2011). All other emissions from fuel combustion in 2005 are calibrated to (Global Emissions EDGAR v4.2, 2013). Emission factors for $SO_2$, BC, and OC are

assumed to decline over time according to air pollution policies based on (Klimont et al., in prep.b). Current near-term policies are enforced in high-income countries, with gradual strengthening of goals over time and gradual technology (Research, Development, Demonstration and Deployment (RDD&D)). Low-income countries do not fully implement near-term policies, but gradually improve over the century.





Emissions from international shipping and aviation and waste of all species are exogenous and taken from (Fujino et al.,
2006). Further, REMIND uses land-use emissions from the MAgPIE model (see section 2.4.1), which in turn are based on
emission factors from (van der Werf et al., 2010).

### 3.4.3 Carbon dioxide removal

In addition to CCS with fossil fuels and in the industry sector, four CDR options are available: afforestation and
reforestation, bioenergy with CCS (BECCS), direct air capture with CCS (DACCS), and enhanced weathering of rocks
(EW). The first two are calculated in the core, while DACCS and EW are calculated in the module "33_CDR".

$CO_2$ emissions from afforestation and reforestation are derived from the land-use optimization model MAgPIE4 (see section
3.2.5). The trade-off between land expansion and yield increases is treated endogenously in the model. BECCS is the only
CDR technology that provides sizable energy instead of consuming it. The idea of BECCS is to turn biomass grown on land
carbon-negative by capturing the emissions arising during combustion or the refinery process. BECCS can be used for
electricity, hydrogen, gas, or liquid fuel production with different carbon capture rates.

DACCS captures $CO_2$ directly from the ambient air. The techno-economic parameterization relies on the literature review
performed in (Broehm et al., 2015). Besides capital investments and O&M costs, DACCS requires heat and electricity. In
REMIND, natural gas or $H_2$ can be used to generate the required heat. There is no explicit limitation to the amount of carbon
removal via DACCS; it is only limited due to costs and the amount of energy and carbon storage that can be provided. EW is
based on the acceleration of the natural weathering of silicate rocks, which is an integral part of the carbon cycle. In
REMIND, those rocks are assumed to be basalt, which is rich in phosphorus and potassium and contains very low
concentrations of trace elements. The basalt has to be mined, ground to small grain sizes, and spread on agricultural fields.
The regional potential for carbon removal depends on the agricultural land and the climate zone as this process is faster in
warm and humid regions and amounts to a maximum of 4.9 Gt $CO_2$/yr removed (Strefler et al., 2018b). Economic costs are
at 200\$/t$CO_2$ removed, including electricity and diesel for grinding and transport. Due to the still large uncertainties
especially in the carbon removal potential, EW is included only in dedicated studies.

In all regions, an additional tax of 50% of the current carbon price is imposed on net-negative emissions to account for
climate damages due to the associated temperature overshoot and governance and finance risks of net-negative emissions.

BECCS, DACCS, and fossil CCS compete for geological storage. Regional annual CCS deployment is limited to 0.5% of
total storage capacity, limiting the total global CCS use to about 20 Gt $CO_2$/yr (values for SSP2, decreased by 50% for SSP1,
increased by 50% for SSP5). To reflect the risk of leakage and the associated possible costs, costs of improved safety criteria
related to monitoring, reporting, and verification, and difficulties due to public acceptance, which are all likely to increase
with deployment, the best estimate of CCS costs is increased linearly such that costs are about 100% or 30\$/tCO2 higher at
maximum deployment.





### 3.5 Representation of other environmental and social impacts

Tackling climate change will not only affect GHG emissions. The deep transformation of the energy system, transportation and industry provides both synergies and trade-offs with broader sustainable development objectives as defined by the UN Sustainable Development Goals (Griggs et al., 2013). As such, IAMs increasingly try to capture additional effects of climate policy, most prominently air pollution (Rao et al., 2016; West et al., 2013; Vandyck et al., 2018; Rauner et al., 2020) and water use (Mouratiadou et al., 2018; Fricko et al., 2016).

REMIND explicitly models the following non-climate environmental outcomes: water withdrawal and usage associated with power generation [module "70_water" (Mouratiadou et al., 2018)] and air pollution emission [module "11_aerosols"], concentrations and human health impacts for all sectors, please refer to (Mouratiadou et al., 2018) and (Rauner et al., 2020) for detailed descriptions of the methodology. Furthermore, environmental and health impacts of the power sector are represented through life-cycle analysis (Luderer et al., 2019; Gibon et al., 2017b; Gibon et al., 2017a), and consequences of mitigation policies for inequality and poverty can be calculated in post-processing. Increasingly, a more comprehensive suite of social and environmental outcomes of climate policy and other sustainability measures is covered (Bertram et al., 2018)), also making use of the interface with the MAgPIE model (Humpenöder et al., 2018).

### 4 Outputs

REMIND provides an integrated view on possible future developments and their implications on the global energy-economy system and explores climate policy options while fully capturing the interactions between economic development, trade, and climate mitigation policies. In this section, model outputs for SSP1, SSP2 and SSP5 scenarios are provided. For each of these assumptions on future development, a scenario with current policy assumptions (NPi) and a climate policy scenario restricting cumulative emissions to a budget of 1300 Gt $CO_2$ (PkBudg1300) and a budget of 900 Gt $CO_2$ (PkBugd900) (counted from 2011, see section 2.5) are shown.

### 4.1 Emissions

Different socio-economic developments feature different strategies to achieve the 1.5°C target (see fig. 5). While $CO_2$ emissions from fossil fuels and industry are reduced by 70-80% in 2050 in all scenarios, the deployment of CCS increases significantly from SSP1 to SSP2 to SSP5. In 2100, the difference is even more pronounced. In an SSP1 setting, $CO_2$ emissions are reduced by 90%, and also $CH_4$ and $N_2O$ emissions from land use are much lower than in the other scenarios due to a lower population growth than in SSP2 and more sustainable lifestyles with less demand for animal-based products. This also leads to less demand for agricultural land and leaves room for regrowth of forests and natural vegetation, thus enhancing the land carbon sink. The SSP5 scenario also assumes lower population growth and therefore sees a similar land carbon sink and lower $CH_4$ and $N_2O$ emissions from land use than the SSP2 scenario. At the same time, it features strong increases in energy demands and relies more strongly on CCS, and therefore does not reduce $CO_2$ emissions significantly in



the second half of the century. In the SSP2 scenario, non-$CO_2$ GHG emissions from land use are hardly reduced and therefore contribute a significant share to the residual emissions in 2050 and 2100.

**Figure 5: Global GHG emissions by type for the 1.5°C scenarios (pkBudg900) based on SSP1, SSP2, and SSP5. The white line**
**shows net GHG emissions.**





While global GHG emissions in the SSP2 scenario are reduced by 15% in 2030, 80% in 2050, and about 100% in 2100, the timing of emission reduction can vary strongly across regions (see fig. 6). In the OECD regions Canada, Australia, New Zealand (CAZ), Europe (EUR), Japan (JPN), non-EU Europe (NEU), and the USA, emissions have peaked already. In most other regions, emissions peak only in 2020 as 2025 already sees strong emission pricing across all sectors. One exception is

India (IND), where emissions only peak in 2025 despite ambitious immediate climate policies. Regional differences are even more pronounced regarding the timing of net-zero emissions. The EU, Japan, and the US reach net-zero emissions at mid-century, closely followed by the Reforming Economies (REF) in 2055, Latin America (LAM) in 2060, and China in 2070. CAZ and NEU achieve emission neutrality only towards the end of the century, and the remaining regions India, Middle-East and North Africa (MEA), Other Asia (OAS), and Sub-Saharan Africa (SSA) retain some residual emissions that are

compensated by net-negative emissions in the other regions.

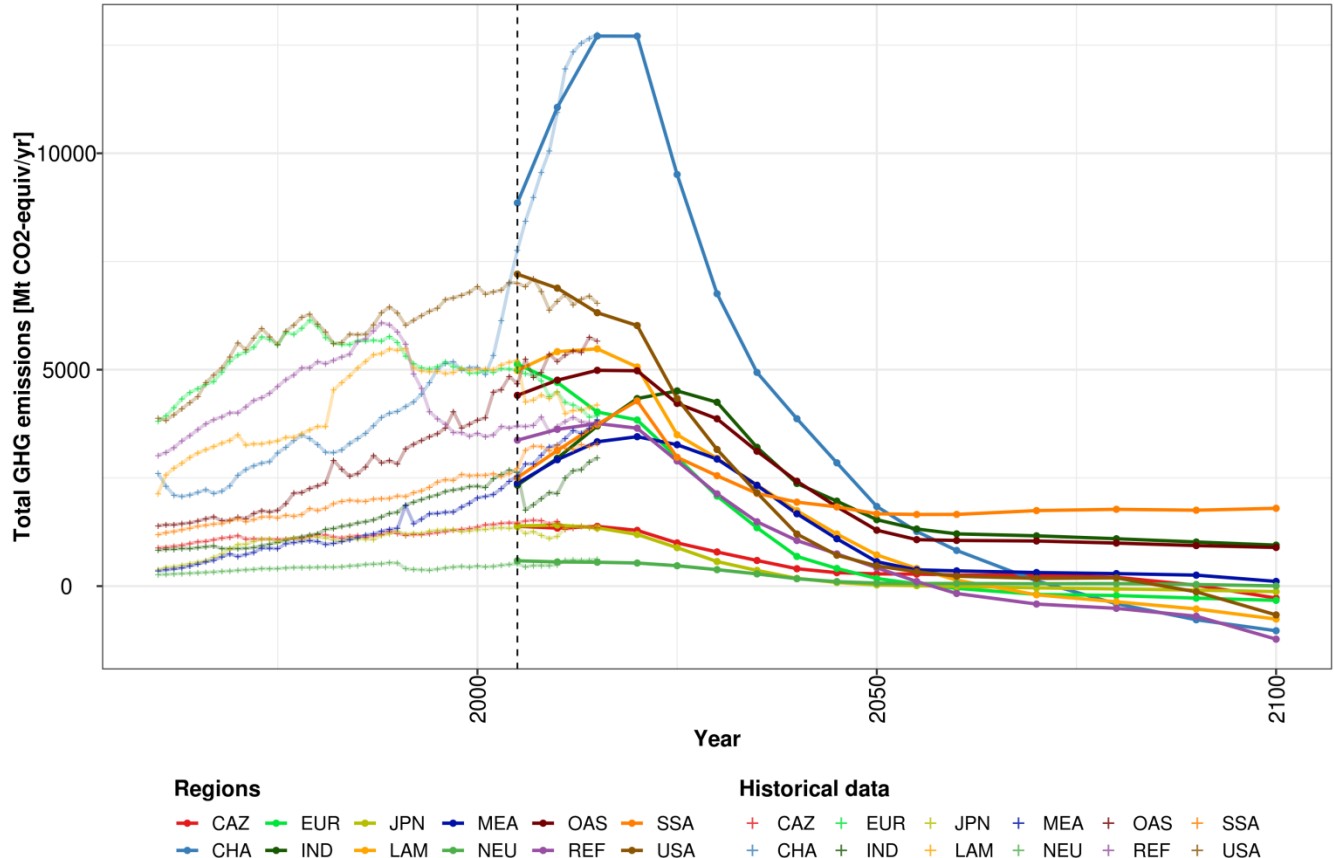

**Figure 6: Regional total GHG emissions by type for the 1.5°C scenario (pkBudg900) of SSP2 with historical data from PRIMAPhist (Gütschow et al., 2016).**



## 4.2 Energy

Socio-economic assumptions as well as climate policy stringency strongly impact the evolution of the global energy system (see fig. 7). In scenarios without strong climate mitigation policies (NPi), fossil fuels will retain a dominant role until 2050. Their dominance would also continue for SSP5 socio-economic assumptions, but would be gradually reduced in SSP2 futures, and would be replaced by a rather diverse energy system with similar contributions from wind, solar, bioenergy and fossils in 2100 in SSP1. The reason for these structural differences are partly due to differing assumptions on technology cost

and resource availability across SSPs, but also due to the main scale effect implying a more than two times higher total energy consumption in SSP5 compared to SSP1. All NPi scenarios project a considerable amount of wind and solar power to be competitive even without ambitious climate policies.

Ambitious climate policies lead to a complete transformation of the global energy system, with most of the transformation already completed until 2050. Coal is quickly phased out completely in the power sector, and only very small residual use

remains in the industry sector (partly enabled by CCS). In SSP1 and 2, oil and gas use is also reduced to very low levels until 2050. In SSP5 however, oil, and especially gas is continued to be used throughout the century, enabled partly by CCS for gas, and by very high levels of carbon dioxide removal (CDR) to offset the considerable residual $CO_2$ emissions from these uses.

Across SSPs, renewables, especially wind and solar dominate decarbonized energy systems, and climate policy in line with

1.5°C results in roughly a doubling of deployment compared to the NPi scenario in each SSP respectively. More importantly still, deployment is accelerated strongly with very high growth rates for both technologies in the coming decades. Very high shares of wind and solar in total primary energy supply in policy scenarios are enabled by stronger and accelerated electrification of all end-use sectors. Nuclear plays no relevant role in climate policy scenarios with SSP1 or SSP2 socio-economic assumptions, but plays an important niche role in the SSP5 variant, where very high electricity demands in some

regions surpass the generation potentials assumed for wind and solar for SSP5. Therefore nuclear power, while not providing larger shares to global electricity production than today, is massively scaled up in such a scenario in absolute terms. The use of biomass, hydro power and geothermal energy is relatively similar across SSP policy scenarios, mainly caused by supply constraints for these options.

The key role of energy efficiency measures for climate mitigation is best illustrated by the reduction of final energy demands

when comparing each of the mitigation scenarios to the corresponding NPi scenario (see fig. 8). The reduction of final energy due to climate policy is strongest in the next few decades when the energy system is in transformation, and is less pronounced once the transformation is completed. As a consequence, the SSP2 and especially SSP5 mitigation scenarios project substantially higher total final energy demands than today for the end of the century, whereas the SSP1 scenario stabilizes FE demand at approximately the current level. In terms of sectoral composition of final energy, neither socio-

economic assumption nor climate policy has a strong impact, with the exception of the noticeable higher share of transport





for the very high final energy demands in SSP5. Mitigation in all sectors involves accelerated electrification (Luderer et al., 2018), though the absolute level of electrification that can be reached varies by sector, and SSP.



**Figure 7: Primary energy mixes by carrier for the NPi and pkBudg900 (1.5°C) scenario of SSP1, SSP2 and SSP5.**





**Figure 8: Final energy mixes for sectors by carrier for the NPi and pkBudg900 (1.5°C) scenario of SSP1, SSP2 and SSP5**

## 5 Discussion and conclusions

Since REMIND is a multi-regional model of the energy-economic system, it is well equipped to capture the interactions
between the energy transformation in response to climate policies and economic development. Full macro-economic
integration is particularly valuable for the assessment of effects of climate policies on the scarcity of energy carriers, demand
response, structural changes, investments, macro-economic costs and their regional distribution.

The central strength of REMIND with its perfect foresight is its ability to calculate first-best mitigation strategies that
provide benchmark development scenarios with detailed representation of the key dynamics related to the scale-up of novel
technologies and integration constraints in the power sector. These benchmark scenarios allow for comparison with
mitigation scenarios under second-best policy settings (regional or sectoral fragmentation) or technology constraints.

Within some numerical restrictions, the flexible spatial resolution of REMIND enables exploring transformation pathways of
the energy-economic system for specific countries or global regions (e.g. Europe).



Due to the simultaneous solution of the macro-economy and the detailed energy system, as well as intertemporal optimization and several non-linear equations in the model, the computational effort for solving REMIND is substantial. This level of computational complexity also puts an upper limit on the amount of detail that can be represented in the model. However, the modular structure of REMIND enables detailed analysis of a specific part of the model (e.g. fossil fuel extraction) tailored to the research question without increasing the numerical burden of the default model. In addition, the feasibility to link REMIND with other models (e.g. EDGE, MAgPIE, MAGICC) guarantees consistent detailed results with small increase of model complexity.

### *Code and data availability*

The REMIND code is available under the GNU Affero General Public License, version 3 (AGPLv3) via GitHub (https://github.com/remindmodel/remind, last access: 1 December 2020, (Luderer et al., 2020a). The technical model documentation is available under https://rse.pik-potsdam.de/doc/remind/2.1.3/ (last access: 1 December 2020) and also archived via Zenodo (Luderer et al., 2020b). The results of the scenarios shown in this paper are archived at Zenodo (https://doi.org/10.5281/zenodo.4313156).

### Appendix A – Comparison with historical data

REMIND generates scenarios which are under no circumstances to be understood as forecasts. It generates possible future projections conditional to specific assumptions which serve as benchmarks (due to perfect foresight and intertemporal optimization) for policy advice. It is not the primary purpose of REMIND, nor any model with a distinct normative component, to reproduce past development. This does not mean that there is no validation of the model. For example, the REMIND model replicates a set of stylized facts of macroeconomic growth and their interrelationship with energy demand (Kriegler et al., 2017). However, the validation criteria are softer and more difficult to define than for purely descriptive and geophysical models. One focus is therefore to match short-term trends. REMIND includes bounds (e.g. capacity of technologies) to emulate the near term future. As pointed out by Schwanitz (2013), validation of IAMs cannot rely to the same extent as for geophysical models on hindcasting, and therefore complementary evaluation approaches such as comparison to more stylized historical trends, or comparison across models are used in addition. A key outcome of transition scenarios is the scale and speed at which new technologies deploy and diffuse. Independent analyses of REMIND scenarios have shown that the model's early periods do not contradict historical experience (Wilson et al., 2013; van Sluisveld et al., 2015). Moreover, the base year calibration of the model, regional energy potentials and the techno-economic assumptions of technologies are regularly reviewed in model comparison studies (e.g. (Luderer et al., 2018; Roelfsema et al., 2020; Bauer et al., 2018; Riahi et al., 2017)).

In the following illustrative results of various REMIND scenarios are compared to historical data. As the model starts in
2005, this demonstrates that results for the overlapping time span 2005-2015(2019) fit to historical data. Future projections take up historical trends and provide plausible results of the future. For population and GDP this is shown in fig. 1 and regional GHG emissions are compared in fig. 6. Fig. 9 demonstrates global primary energy pathways for coal, oil, gas and biomass compared to historical data from IEA. Trajectories of global total final energy and final energies of the sectors buildings, industry and transport in comparison to IEA data are shown in fig. 10.

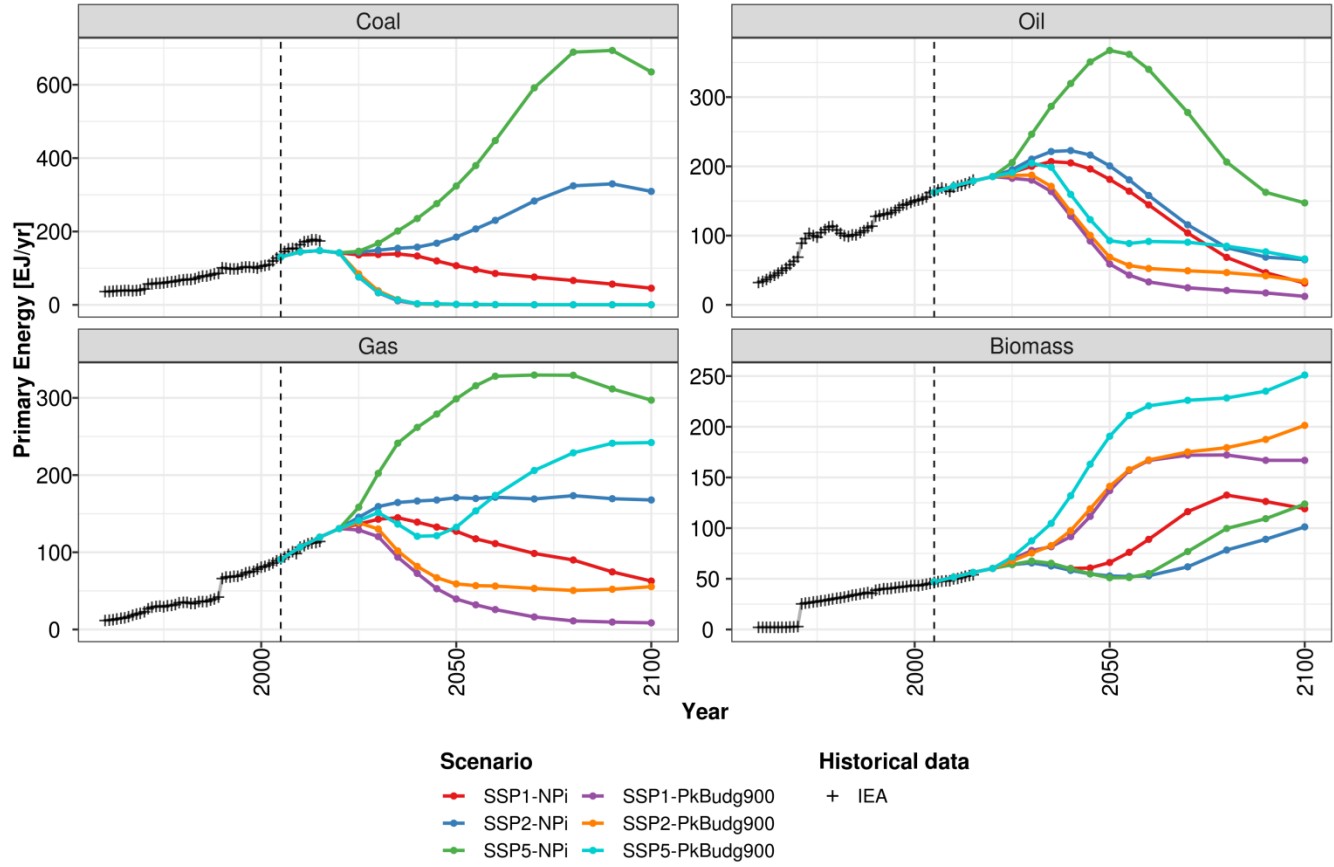

**Figure 9: Global primary energy consumption of different energy carriers for NPi ad pkBudg900 (1.5°C) scenarios of SSP1, SSP2 and SSP5 compared to historical data from IEA**

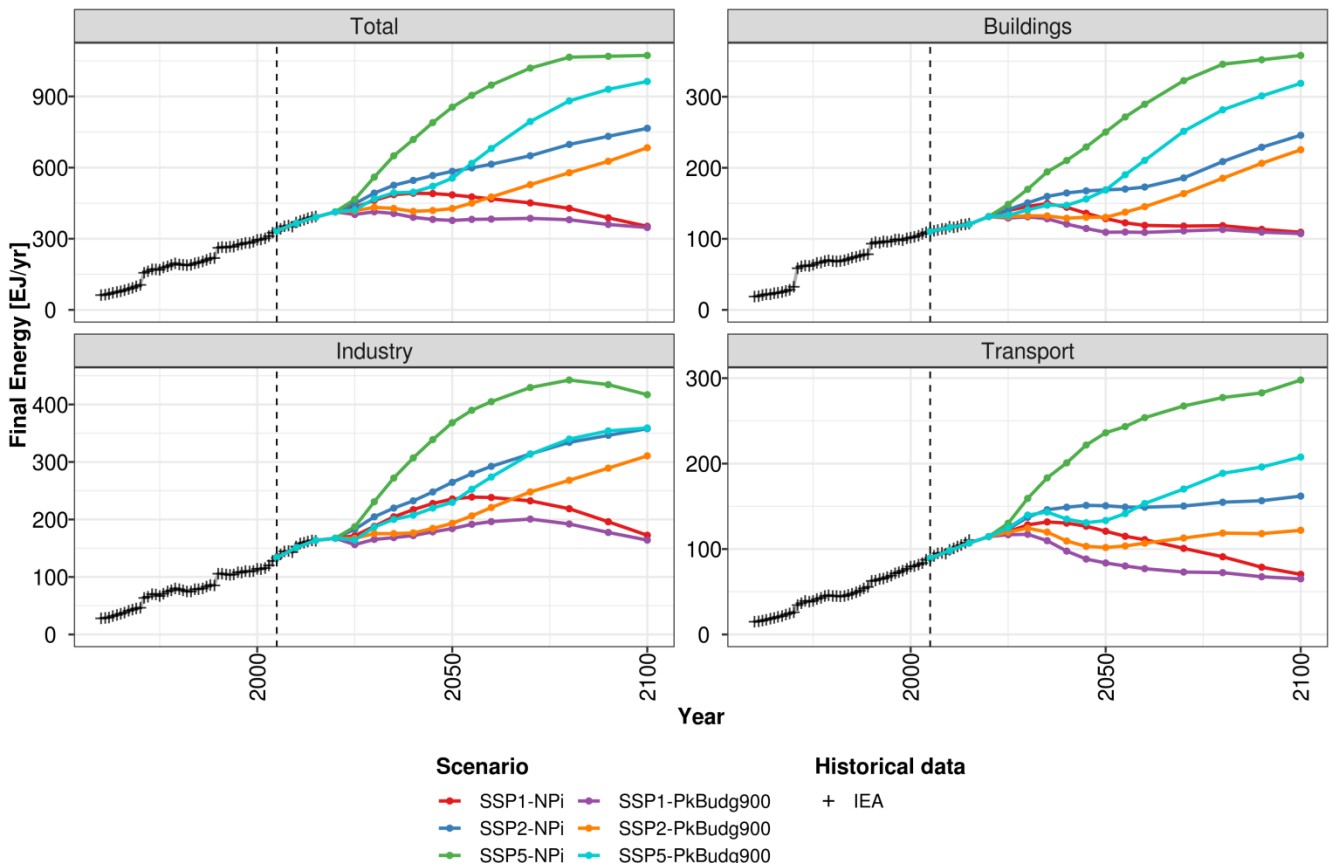

**Figure 10: Global final energy for sectors for NPi and pkBudg900 (1.5°C) scenarios of SSP1, SSP2 and SSP5 compared to historical data from IEA**

*Author contribution*

GL and EK supervised the development of the model regarding content. LB, AG, DK and JPD provided technical support to the development of the model framework. LB performed the simulations and prepared the manuscript with contributions of NB, CB, DK, JK, ML, AL, SM, MP, RP, FP, SR, RR, MR, JS, FU and GL. LB, DK and FB created the figures shown in this paper. All authors contributed to the development of the model framework and the manuscript.

*Competing interests*

The authors declare that they have no conflict of interest.



*Acknowledgements*

The research leading to these results has received funding by the German Federal Ministry of Education and Research (BMBF) under the grant agreement number 03EK3046A (START project) and under the grant agreement number 03SFK5A (ARIADNE project). This work was also supported by the German Research Foundation (DFG) Priority Programme (SPP)
(CEMICS2 projects) and by the European Union's Horizon 2020 research and innovation programme under grant agreement numbers 821124 ( NAVIGATE project) and 821471 (ENGAGE project).

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
