# Peer review of "REMIND2.1: Transformation and innovation dynamics of the energyeconomic system within climate and sustainability limits"

_Geoscientific Model Development, 2021_

## Author Comment (AC5)

Dear topical editor,

Thank you for your helpful comments, shown in italics below. Find our replies directly below each comment.

*In addition to the comments by the reviewers and the executive editor, please also take into account the following considerations when preparing the submission of responses and an updated version of the manuscript:*

*- There are references to the code in the style "number plus module name" throughout the manuscript. It is not clear what the logic of the numbering is, so I'd suggest to provide an explanation and/or an overview figure or table with all (or at least the most relevant and referenced) modules.*

A description of the modular structure of REMIND will be included in the revised version of the manuscript including an explanation of the style of our module realizations "number plus module name".

*- In the section on steady-stage and equilibrium, you should introduce the general-equilibrium concept early on (not in the last paragraph). Also, this section should cross-reference the "perfect-foresight" assumption of REMIND.*

Section 3.1.2 will be revised accordingly.

*- Table 2 does not have a very complicated structure and could be replaced by a sentence or a list.*

We agree that table 2 clearly could be replaced by a sentence or a list but we prefer a table with the advantage that it is much easier to grasp.

*- The "additional tax of 50% of the current carbon price" on net-negative CO2 emissions (page 29, line 777f) seems to be a very arbitrary modelling choice. Please provide a rationale for this value.*

The rationale for the additional tax of 50% of the current carbon price on net-negative emissions is two-fold: firstly, as soon as total emissions turn net-negative, carbon pricing is not providing revenues anymore but requires net government spending. Secondly, geophysical reasons speak for rather limiting overshooting cumulative emissions budget. The 50% is simply the middle ground between treating net-negative emissions the same as emission reductions or not allowing for net-negative emissions at all, i.e. a tax of 100% which would remove all revenues. We added this motivation to the revised version of the manuscript.

*- The term "internally consistent" may be more intuitive than "self-consistent".*

The term "self-consistent" is changed to "internally consistent" in the revised version of the manuscript.

*- The phrase "investments turn out regrettable" (p13, line 337) and "capital is enlarged" (page 18, line 465) should be revised.*

The first sentence is changed to "As prices change earlier, it turns out that some investments went in the wrong direction (e.g. wrong technology) and adjustments are made in the next period." The second sentence is changed to "Investments increase capital stocks which depreciate according to the depreciation rate...".

*- The sentence "the marginal of the (variable of) taxed activities is impacted by the tax [..]" (page 19, line 499) is not clear.*

We have changed this sentence to "Nevertheless, the marginal of the variable (but not the parameter) of taxed activities is reflecting the tax rate which leads to the intended adjustment effect in the CONOPT solution.".

*- The phrase "these assets are then stranded" (page 23, line 573) should be revised.*

We have changed this sentence to "Capacities are phased out before they reach the end of their technical life-time by the optimization if the value of their outputs is lower than the costs of variable inputs, reflecting a situation of asset stranding. This happens especially in 'delayed' scenarios, which start optimization only at a future point in time."

*- Subsection header 3.4 should be renamed, as this section also includes non-GHG emissions.*

We have changed the header of subsection 3.4 to "Representation of emissions".

*- "roughly a doubling" (page 33, line 850) should be revised.*

We changed the sentence to "... and climate policy in line with 1.5°C results in twice as much deployment compared to …"

*- Section 5 is quite short and the section title "Discussion" is therefore not adequate.*

The title of section 5 will be changed to "Conclusions" in the revised version of the manuscript.

*- The programming language(s) should be clearly stated in the section "Code and data availability".*

We added the sentence "The REMIND code is implemented in GAMS and code and data management uses R." to the section "Code and data availability" in the revised version of the manuscript.

*- Please use the year of the latest update of the model description when citing the IAMC wiki (currently, it does not have a year in the reference)*

We corrected the reference.

---

## Author Response (AR1)

REFEREE #1:

Dear referee,

Thank you for your helpful comments, shown in blue and italics below. Find our replies directly below each comment.

*The manuscript presents a detail description of the REMIND model. It is well written the content is mostly very clear. However, there are some comments that must be addressed before the manuscript could be considered for publication. The main concern is that it is not clear what new knowledge about REMIND this manuscript brings to the literature. There are several other research papers that describe several components of REMIND (cited in this manuscript as well) that should be better described here (not just refer the reader to other papers).*

This manuscript introduces the new version 2.1 of REMIND. The last comprehensive documentation of REMIND described version 1.7. In the meantime, REMIND has improved substantially and become open source. Therefore, we would like to provide a complete description of the model which fills in all the missing information and interlinkages which are not included in previous publications focusing on specific aspects of the model.

*Additionally, it is not clear what is new in this version of REMIND, compared to previous versions that have been published. The manuscript is not clear about this and does not present a comparison or a section that explicitly presents the improvements.*

We added section 1.3 on "What is new in REMIND 2.1?" to the revised version of the manuscript, containing the following points, was added:

- flexible spatial aggregation for input data generation
- open source
- update of techno-economic parameters for most technologies to reflect latest market data
- updated bounds on developments until 2019 to reflect latest deployment and policy developments
- updated policy scenarios
- more detailed representation of demand sectors buildings, transport and industry
- possibility to include aggregated representation of impacts
- possibility of imperfect capital markets

*Other minor/detail comments are presented next.*

*Comments Section 1:*

*Main comment: It is not clear from this section what is new in this version of REMIND to respect the older versions. This needs to be clearly stated. Other minor comments are described next:*

 1. *The use of SSPs helps to cover uncertainties regarding technological development for renewable or fossil fuel availability, but also social and behavioral development like*

*population growth, dietary preferences, environmental awareness or international cooperation.*

This sentence was changed slightly: "...uncertainties regarding technological development for renewable energy or fossil fuel availability, but also social and behavioral developments like…"

2. *It is not clear what is meat by "damages" in the following text:* **, the REMIND model represents some damages and can thus be used for cost-benefit analyses or least total cost analyses**

In REMIND, "damages" refer to economic damages due to climate change. This sentence was changed to "REMIND can endogenously represent macro-economic climate change damages based on recent damage function estimates (Kalkuhl and Wenz 2020)" in the revised version of the manuscript.

3. *In the next sentence, what is meant by self-consistent? Also, it is understood that the energy sector is modeled, while the economic (macroeconomic) is input. Hence, it seems a bit contradictory the next sentence: REMIND enables the exploration of a wide range of plausible developments and of possible futures of the **energy-economic** system exploring **self-consistent** transformation pathways.*

The SSP scenarios provide a wide range of possible macroeconomic developments. While the population in REMIND is exogenous information and REMIND is calibrated for baseline scenarios to match GDP and final energy trajectories, investment decisions into the macro economic capital stock and energy carriers are endogenous. In particular, changes in macroeconomic variables from baseline scenarios to policy scenarios are endogenous. In the revised version of the manuscript we substituted "self-consistent" with "internally consistent", and the ensuing sentence explains that this refers to the simultaneous consideration of various sectors, including a simplified macro-economy, and the interplay of technology transformation and socio-economic changes.

4. *It is mentioned several times (up to page 3) that REMIND is an NLP, however, no description or ideas of the non-linear components are described. What make the model nonlinear? Some initial hints would be beneficial .*

In REMIND several model components are represented via nonlinear equations (e.g. welfare function, CES production function, technological learning curve). We added this information in the corresponding sections of the revised manuscript. In Section 3.1.1 we adjusted the sentence to: "Social welfare is equal to the discounted intertemporal sum of utility, which itself is a nonlinear function of per capita consumption." and in section 3.1.3 we adjusted the sentence to: "The production function represents a system of nonlinear equations or more specifically a nested CES (constant elasticity of substitution) function with capital, labour, and final energy as inputs."

5. *"REMIND calculates economic and energy investments". What is meant by an "economic investment"?*

By "economic investments" we mean investments into the macro-economic capital stock (i.e. physical capital that represents machinery, long-lived equipment, infrastructure, etc. beyond the core energy system represented explicitly via separate technologies). We changed this sentence to "...calculates aggregate macro-economic as well as technology-specific energy-related investments…".

*Introduce the concept of Pareto optimum. Pareto is normally use in the context of multi-objective optimization.*

We apply the concept of Pareto optimality in an economic context (cf. Mas Collell et. al., 1995, p. 312 ff), specifically in a setting where different regions interact with each other. A solution is Pareto optimal if there is no other allocation of economic income and resources that would increase the welfare of one region without decreasing the welfare of another. By using the Negishi approach, REMIND directly finds a Pareto optimum that is, by construction, equivalent to the competitive equilibrium found by the Nash approach with full internalization of externalities. We added an appropriate explanation in paragraph 5 of section 2.2.

7. *"The optimization is subject to equilibrium constraints, such as energy balances, economic production functions or the budget constraint of the representative household." It is not clear the mathematical structure of the REMIND model. Is it an NLP in the form of an optimization or equilibrium (MPEC) model? (MPEC: Mathematical program with equilibrium constraints).*

The model is formulated as a standard nonlinear programming model with an objective function and a large number of constraints that represent different parts of the overall system. These constraints represent equilibrium or balance conditions, but are not explicitly differentiated as such. This way of model formulation is less strict than the MPEC models. This flexibility is needed as the model merges together different components of the energy-economy-climate system that are quite different in nature. We changed the beginning of section 2.2 to "REMIND, as a composition of different modules and components, is mathematically coded as a nonlinear programming model, i.e. a model with a single objective function and a large number of side constraints. As such, it is computed by the solver CONOPT which is supposed to find a local optimal solution.".

8. *"REMIND is usually run in a decentralized mode where each model region is optimized separately, and clearing of global trade markets ensured via iterative solutions (see section 2.2)." -> How the model ensures convergence? This needs to be clarified in the corresponding section.*

In order to resolve trade interactions between model regions (i.e. clearing markets), the decentralized (Nash) approach applies a Walrasian type price adjustment algorithm. Regional actors start from an initial price vector and choose their trade patterns, acting as price takers. The regional solutions are subsequently collected, and the price for the next iteration is adjusted based on the surplus and deficits on the markets. Walrasian-type price adjustment algorithms are commonly used and convergence is conceptually proven under generous conditions (see also section 3.1.2). The implemented specification of the price adjustment algorithm (see details in

Leimbach et al., 2017) makes use of parameters that play the role of price elasticities and help the model to converge. In order to guarantee convergence, two auxiliary mechanisms are applied: (i) anticipation of price changes, and (ii) penalty costs depending on the change of regional trade patterns over iterations. We extended paragraph 3 in section 2.2 to provide this information.

9. *"CH4 emissions from fossil fuel extraction and residential energy use" What about CH4 from the agro-industrial (food, beef-lamb) sectors? If not considered, it must be clarified.*

CH4 emissions from the land-use sector as well as from waste handling are considered. However, they are only calculated via emission baselines and marginal abatement cost curves (MACCs) - not directly by source. We reformulated the text to mention CH4 emissions from fossil fuel extraction and residential energy use under section 3.4.1 together with the GHG emissions from other sources.

10. *"Historical data for the year 2005 is used to calibrate most of the free variables (e.g. primary 130 energy mixes in 2005, secondary energy mixes in 2005, standing capacities in 2005, trade in all traded goods for 2005)." Why there are not updates to the base calibration year? 2005 seems a bit old to account for new trends.*

We calibrate variables to 2005 in order to have some years of overlap between model results and historic values, which are useful to confirm that REMIND can replicate observed trends. Significant departures from near-term developments are addressed by applying some bounds, e.g. on technology availability and trade volume. We added the sentences: "Additional bounds for a select few variables, primarily capacity (additions), up through 2019 ensure that the 2020 point of departure in current policy cases is proximal to actual developments. The ability to also run the simulation without these constraints enables important comparisons of model dynamics from 2005-2020 with real-world developments."

**Comments Section 2:**

1. *"(for more information about the modular structure see Dietrich et al., 2019 - Appendix A." It would be good to introduce some of this information in this article, since it is such a critical piece of the model structure of REMIND.*

We added Appendix C and the following paragraph in section 2: "The name of each module starts with a two-digit number. Each parameter and variable of the REMIND code follows a naming convention: a prefix first indicates the type of object (e.g. "v" for variables and "p" for parameters), and second whether it is only used inside one module (i.e. using the module number) or as an interface with at least one other module or the core (i.e. using "m"). For example, the variable "vm_taxrev" is an interface between the module "21_tax" and the core, while the variable "v21_taxrevGHG" is only used inside the module "21_tax". Appendix C gives an overview of all modules used in REMIND. "

2. *"This paper focuses on realizations which are active in default scenarios. More detail about all modules and their interlinkages can be found in the model documentation". I*

*still believe that this is information is relevant and should be somehow described and discussed in this manuscript.*

The module structure in general enables a clear structure of the code and splits it into meaningful components with clear interfaces. A bit more explanation about this module structure will be added in the revised version of the manuscript. Although the focus of this manuscript is on the default realizations of a module, in most cases a short description of alternative realizations is given (e.g. description of the damage module in section 3.1.5). Previous publications which focus on the more detailed realizations are often mentioned as well. In some cases there exists only one realization of a module. This is indicated in the revised version of the manuscript for the following modules: "01_macro", "02_welfare", "04_PE_FE_parameters", "05_initialCap", "11_aerosols". Nevertheless, a complete list of all available module realizations is provided by our model documentation.

3. *"By default REMIND calculates results for the 12 following world regions:" A table with regions and other important information would be better than just listing countries/regions.*

We added Appendix B containing a table of regions and countries belonging to those regions in the revised version of the manuscript.

4. *"By parallelizing the calculation of the individual regions in decentralized optimization mode (see section 2.2) the computation time increases only moderately with increasing spatial detail." It would be interesting to have a general idea of the computational complexity of the model (minutes, hours, days?) depending on the type of scenarios.*

REMIND converges within 1 to 3 hours for scenarios with the default settings. If some modules are used in a more complex version, this can increase to 5-8 hours. The runtime only slightly varies with policy type, with intermediate ambition policy scenarios without many policy-specific constraints having longest runtimes. In line 199 of the first manuscript, we provided the rough estimate: "(due to the possibility of parallel computing, scenarios converge within one to a few hours)". We now specify further: "both baseline and policy scenarios converge within one to a few hours, mostly depending on the specified module detail."

5. *"Time represents a separate dimension" -> What is meant by this? Not clear at all.*

In economics, static models are widely used. They just analyse economic activities in a given point of time and the model dimension is mainly determined by the number of actors (firms, consumers) and goods (sectors, markets). In a dynamic model like REMIND, the dimensionality, and therefore the numerical demand, is largely influenced by the time dimension, i.e. time horizon and time steps. We specified further: "Time represents a separate dimension within all equations - alongside the also ubiquitous spatial dimension and further equation-specific dimensions relating to technologies, emission species, etc. - increasing the overall dimensionality of the model."

6. *"In essence, the time dimension only increases the number of markets for which the algorithm has to find an equilibrium" I would be extremely careful about the use of the concept "equilibrium". To this point, the model has been introduced as a NLP optimization problem, with some equilibrium constraints. But there is not clear mathematical structure to really understand what the model does. If it is a pure optimization model, what talk about equilibrium? Why Pareto optimal is mentioned earlier? Please be clear and consistent with the type of solution that is obtained.*

There is a clear context in which we use the concept of equilibrium: trade interactions of model regions. As part of the model solution, an equilibrium is found that is characterized by a set of prices for tradable goods that clears all markets. While we do not prove the existence of this equilibrium for the model, we apply algorithms that are known to generate an equilibrium: Walrasian-type price adjustment.

A corresponding context is given for the concept of Pareto optimality. Here, we repeat our response to question 5 in section 1: We apply the concept of Pareto optimality in an economic context (cf. Mas Collell et. al., 1995, p. 312 ff), specifically in a setting where different regions interact with each other. A solution is Pareto optimal if there is no other allocation of economic income and resources that would increase the welfare of one region without decreasing the welfare of another. By using the Negishi approach, REMIND directly finds a Pareto optimum that is, by construction, equivalent to the competitive equilibrium found by the Nash approach with full internalization of externalities. We added an explanation in paragraph 3 and paragraph 5 of section 2.2.

*Also, it was mentioned that CONOPT is used to solve the NLP problem, hence, it is also questionable when authors refer to the "algorithm" use to solve REMIND, since it is in fact a solver who does this process and authors have not developed an algorithm. In the case that an algorithm is indeed implemented, then this has not been clearly stated and differentiated from the NLP-CONOPT process.*

In the first two paragraphs of section 2.2 we adjusted the manuscript to explain in more detail the two-layer structure of the solution process in REMIND (inner and outer optimization). While we did not develop algorithms for the inner optimization (using the NLP solver CONOPT), we developed algorithms for the outer optimization that we use in order to generate solutions that are meaningful from an economic point of view (and that make use of corresponding economic concepts).

7. *Based on the previous comment, I found then that there is indeed a NASH mode in REMIND. This helps to understand the concept of equilibrium. However, there is still not clarity in terms of what the base structure of REMIND is, how different structures are solve, what type of solution is obtained, solution algorithm, etc. This needs to be further clarified.*

REMIND, as a composition of different modules and components, is mathematically coded as a nonlinear programming model. As such, it is computed by the solver CONOPT. Basic features of the underlying solution algorithm are hidden. There is however a second layer of solution structure. This is related to algorithms that we use in order to generate solutions that are meaningful from an economic point of view (i.e. Pareto optimum and competitive equilibrium, respectively). In order to find such

equilibrium solutions with the REMIND model, trade interactions of regions have to be reconciled. Two corresponding mechanisms are implemented: Nash and Negishi algorithm. Both are iterative algorithms. Section 2.2 is revised in order to provide more clarity to the different solution structures and algorithms. Within each iteration, the entire NLP model (which itself is solved within thousands of iterations) is computed. The Nash and Negishi approach are briefly described in section 2.2.

*Comments Section 3:*

1. *"It is possible to compute the Pareto-optimal global equilibrium including inter-regional trade as the global social optimum using the Negishi method (Negishi, 1972), or the decentralized market solution among regions using the Nash concept (Leimbach et al., 2017)". This is interesting but needs further clarification. In practice, a Nash solution is an equilibrium, that can be categorized in some conditions as Pareto optimal. In fact, it has been studied that Pareto optimal strategies are a subset of Nash Equilibrium strategies (see paper DOI: 1109/ICCCNT45670.2019.8944817)*

   See reply to comment 6, section 2.

2. *"REMIND considers the trade of coal, gas, oil, biomass, uranium, the composite good (aggregated output of the macroeconomic system), and emissions permits (in the case of emissions-trading-system (ETS) based climate policy).". Are ETS global in REMIND? Or can be defined for particular regions? If global, how are allowances distributed?*

   The default climate policy implementation is via carbon prices, which can either be prescribed exogenously, or can be iteratively adjusted to achieve a prescribed near-term emission target or long-term cumulative carbon budget. There is, however, also the option of defining a global emission budget with various alternative prescribed distributions of permits which then can be traded across regions. This option has been used most recently in Leimbach and Giannousakis 2019 (https://link.springer.com/article/10.1007/s10584-019-02469-8). In response to your comment, we changed the text in parentheses to "(in the case of emission-trading-system (ETS) based climate policy, which is not the default but has been used in some studies, most recently in Leimbach and Giannousakis 2019)".

3. *"To match 2005 values in the IEA statistics, REMIND adjusts the regional by-production coefficients of combined heat and power (CHP) technologies." This refers to the calibration of REMIND in the Energy Sector? If so, it is still not clear how the full calibration process works since it will also depend on the macroeconomic results and other sectors that may not be linked to CHP plants (transport for instance?).*

   Yes, this sentence refers to the energy transformation part of REMIND. What we mean by this sentence is that in different world regions, CHP plants have different electricity to heat output ratios according to the IEA statistics. We reformulated this paragraph to make clearer that module "04_PE_FE_parameters" calculates regional conversion efficiencies and CHP coefficients from IEA energy statistics, and module "05_initialCap" calculates the vintage capital stock needed to produce and convert these amounts of energy.

4. *"represent challenges and options related to the temporal and spatial variability of wind and solar power" Please elaborate on this. If the mode runs with 5 years' time steps, how the temporal variability is considered? It is not inter-annual? Curtailment rates, which are mentioned later, will also depend on the increased levels of demand in future periods as well as the inclusion of other flexibility technologies, such as electrolyzer, which can transform excess electricity into a different energy carrier. Hence, it does not seem correct to consider curtailment rates.*

Bridging the time scales of investments (5 year time steps) and power sector variability (inter-annual) has been one research focus of our group over the past 10 years. In order to make clearer how we have a parametric representation of variability (which functions on time scales of hours and days) in our model with 5-year time steps, we changed the sentence "These drivers are parameterized for a range of wind and solar PV generation shares, as well as for the regional-specific temporal matching of electricity demand and renewable supply." to "These variables are linked via specific equations to the shares of VRE generation, with higher VRE shares resulting in higher requirements for storage and grid. The parametrization of these equations also takes into account the region-specific temporal and spatial matching of electricity demand and renewable supply, so that regions with better concurrence (e.g. large noon demand peaks for air conditioning) require less storage, and regions with higher geographical proximity of VRE resource and demand require less grid investment."

REFEREE #2:

Dear referee,

Thank you for your helpful comments, shown in blue and italics below. Find our replies directly below each comment.

*Overall I found the paper to be well-written and to provide a reasonably comprehensive review of the different model components. I thought there was an appopriate level of technical descriptions, with links to references where individual topics are discussed in greater detail. At a high level, my biggest question pertains to the purpose of this paper in the peer reviewed literature; i.e., can the authors state in the text what the value added is of this paper (as opposed to the model)? As noted, there are already several published model documentation papers, as well as reasonably comprehensive online model documentation. Similarly, the results shown in this manuscript were a cursory review of SSP scenarios that were already published and documented in a number of papers four years ago. It doesn't seem appropriate to be re-publishing this scenario data as if it were new.*

The main purpose of this manuscript is to provide a comprehensive description of the new version 2.1 of REMIND. The last comprehensive documentation of REMIND described version 1.7. In the meantime, REMIND has improved substantially and become open source. Therefore, we would like to provide a complete description of the model which fills in all the missing information and interlinkages which are not included in previous publications focusing on specific aspects of the model. Now we can provide an important update of the SSP scenarios, serving as example results to introduce this new version of REMIND. In section 4 we adjusted the manuscript to make clear that this is an update of the previous SSP scenarios derived by REMIND 1.6, also pointing out that the scenarios reflect changes in systems representation and spatial resolution, but also are an update in that policy scenarios only start to diverge from 2020 onwards, instead of after 2010 for the original SSPs.

*One way that the authors could differentiate this from the prior literature would be to run scenarios that illustrate the value of new features that have been added to the model since the last documentation in 2017. For example if there's more sophisticated representations of variable renewable energy, the paper could show energy curtailment by region and scenario, or other variables that are interesting but that weren't reported in the SSP inter-comparison exercise and perhaps weren't available at that time anyway.*

We added section 1.3 on "What is new in REMIND 2.1?" to the revised version of the manuscript, containing the following points:

- flexible spatial aggregation for input data generation
- open source
- update of techno-economic parameters for most technologies to reflect latest market data
- updated bounds on developments until 2019 to reflect latest deployment and policy developments
- updated policy scenarios
- more detailed representation of demand sectors buildings, transport and industry
- possibility to include aggregated representation of impacts

- possibility of imperfect capital markets

*In terms of reviewing the model, I was generally impressed by the number of features and key interactions captured, but noted two weaknesses that should be explained in the text. First, why is the model calibration year 2005 when it is currently 2021, and the necessary data to calibrate the model to more recent years has been available for a long time? I'd think that the calibration year should be 2010 at a minimum.*

We calibrate variables to 2005 in order to have some years of overlap between model results and historic values, which are useful to confirm that REMIND can replicate observed trends. Significant departures from near-term developments are addressed by applying some bounds, e.g. on technology availability and trade volume. We added the sentences: "Additional bounds for a select few variables, primarily capacity (additions), up through 2019 ensure that the 2020 point of departure in current policy cases is proximal to actual developments. The ability to also run the simulation without these constraints enables important comparisons of model dynamics from 2005-2020 with real-world developments."

*And, second, why are there only 12 global regions? For policy modeling purposes it's often advantageous to have single-country regions, and 4 of the 12 are single-country which is good, but that leaves some very heterogeneous regions. Canada-Australia/NZ seems an especially interesting market region given that they're at opposite sides of the world. Is there any sub-regionalization in the renewable energy markets, or any other way to prevent windy regions of Canada from supplying electricity to buildings in New Zealand? Similarly, "Other Asia" presumably includes a very wide range of development levels, as South Korea is mixed in with a large number of low-income countries. Can the authors comment on the level of effort/difficulty with adding regions to the model? Perhaps several components already include enhanced detail?*

The spatial aggregation of REMIND is flexible: input data can be provided in any spatial aggregation of countries and the model code is automatically adjusted based on a mapping which defines the spatial aggregation. In general, there is a limit to the number of regions due to the solution algorithm, but also due to the effort required for validating results especially for smaller countries/regions. Given the advent of parallelization in the Nash solution mode, the runtime does not increase substantially with higher numbers of regions. However, we only have limited experience as to the number of regions that the algorithms for market clearing can handle and still return a stringent market clearing solution. The 12-region spatial aggregation is considered REMIND's default, for which we have validated the input data and model results. Each additional country/region that is modeled explicitly needs to be validated against detailed historical data. The first steps for increasing spatial detail -validating a version where Europe is split into 11 sub-regions - are currently underway. Further spatial detail is possible but would require validation of the model output. Therefore, for the time being, we have to live with the artefacts mentioned by the referee. They however are not too problematic on the global scale: CAN, AUS and NZ each have good renewable resources, and rather limited populations. The demand estimation in many of the sub-modules calculates demands on country level, and aggregates to the region level, so that some of the heterogeneity within a region is partly reflected in the regional parametrization.

The final thing I was wondering about the model pertains to the renewable energy supply curves, which appear to include uninhabited lands of Russia, Canada, Australia, etc. Am I correct in understanding that all land area is included in these supply curves, starting in the base year, and that there's no consideration of transmission line distance? While this would be a difficult thing to do well (chicken and egg issue with the transmission lines), that does seem a pretty major omission that would tend to make much more wind energy available for much cheaper that it should, for countries like those named above that have large tracts of uninhabited land, thousands of kilometers from any population centers.

The renewable potentials we use include only areas with a maximum of 100km distance to existing settlements. Accordingly, we potentially slightly underestimate the renewable potential for large regions with low population density. However, as these regions do not have a scarcity of wind and solar supply, increasing the renewable potential would not have any impact on our results.

Furthermore, the parametrization of grid demands (which are a function of wind share in power generation) takes this into account, at least in an approximate way. In a recent paper soon to be published, we see that REMIND rather even overestimates the grid investment requirements for integrating solar and wind, at least in comparison with estimates from other models and the IEA.

What follows are some minor questions and requests for clarification:

* How is proprietary data masked or filtered and re-processed for distribution, given that the model is open source but (presumably) not all data used in its calibration is free?

Our strategy for proprietary input data distribution is to release the necessary excerpts at REMIND's default level of regional, sectoral and temporal aggregation, so that the comprehensive proprietary source data doesn't need to be published. We have contacted all of our data providing institutions for approval, but are still awaiting an official agreement from the last one (IEA). In the meantime, our fully open-source data processing routines (R packages) help users to generate input data locally, but the user must have access (a license) to the raw data at the moment.

* Line 34 - should be "example", not "exemplary".

This is changed in the revised version of the manuscript.

* Fig 1 - I believe "labor efficiency" should be re-named "labor productivity" for consistency with the literature.

We adjusted the figure in the revised version of the manuscript.

* Line 166 (and others): The China region is called "CHA" on line 166 and "CHN" on line 170. My preference would be to always use CHN, the official 3-digit ISO code, similar to the handling of the other single-country model regions (USA, IND, and JPN).

The region "CHA" of the default regional aggregation of REMIND contains China (CHN), Hong Kong (HKG),Taiwan (TWN) and Maccao (MAC). Because of this we do not use the official 3-digit ISO code for China for our region (which we used in the previous REMIND versions that

only mapped CHN to the region). This is clarified in the revised version of the manuscript in Appendix B, which shows the region mapping. Regarding macro-economic development and climate policy mitigation, we assume that our region "CHA" is dominated by China. We adjusted the regional description in the text to "CHA - mainly China" to make it clear that "CHA" contains more countries than just China.

*Can the authors provide a country-to-model-region mapping list in an Appendix? A number of the boundaries are unclear from the descriptions (e.g., Latvia/Estonia/Lithuania, Turkey).*

We added Appendix B containing a table of regions and countries belonging to those regions in the revised version of the manuscript.

*Line 286 - should be "modes", not "models" (I think; please check)*

This is changed in the revised version of the manuscript.

*Line 615 (about hydropower potential): "The regional disaggregation is based on information from a background paper produced for this report (Horlacher, 2003)" I'm wondering if this is a typo, or perhaps copied from an older document? Otherwise I can't see how a paper published 18 years ago was produced for this report.*

Please excuse our formulation, which could be misunderstood. What we wanted to say is that there is a 2003 report that states global technological hydro potentials, and a 2003 background paper to that 2003 report that provides regional detail. We reformulated to "These estimates are based on the technological potentials provided in the report (WGBU, 2003) and the background paper produced for this report (Horlacher, 2003)"

*Lines 700-715: for the more detailed version of the buildings module, can the authors comment on how this was calibrated? The disaggregation of energy consumption to the services is not something readily available in external data sources, and the paper cited is under review so a brief description here would help.*

As for the simple realisation, the calibration of the REMIND buildings module is based on the EDGE-Buildings projections. EDGE-Buildings disaggregates IEA Energy balances by end-uses in accordance with additional datasets, and can therefore project energy demand for end-uses as well as energy carriers. The methodology is described in detail in the paper (Levesque et al., 2018, https://doi.org/10.1016/j.energy.2018.01.139). An explanation is added in section 2.4.2 relating to the buildings calibration ("EDGE-Buildings projections are disaggregated both by energy carrier as well as by energy service and can therefore be used to calibrate the different buildings module realizations (see section 3.3.2"). The paper cited in the section describing the buildings module is now published (https://doi.org/10.1088/1748-9326/abdf07), and the citation will be modified accordingly.

**TOPICAL EDITOR:**

Dear topical editor,

Thank you for your helpful comments, shown in blue and italics below. Find our replies directly below each comment.

*In addition to the comments by the reviewers and the executive editor, please also take into account the following considerations when preparing the submission of responses and an updated version of the manuscript:*

*- There are references to the code in the style "number plus module name" throughout the manuscript. It is not clear what the logic of the numbering is, so I'd suggest to provide an explanation and/or an overview figure or table with all (or at least the most relevant and referenced) modules.*

A description of the modular structure of REMIND is included in the revised version of the manuscript in section 2 including an explanation of the style of our module realizations "number plus module name". We also added Appendix C providing an overview of the most important modules used in REMIND.

*- In the section on steady-stage and equilibrium, you should introduce the general-equilibrium concept early on (not in the last paragraph). Also, this section should cross-reference the "perfect-foresight" assumption of REMIND.*

We introduced the general-equilibrium concept in paragraph 3 of section 3.1.2. and refer to the assumption of perfect information in paragraph 4.

*- Table 2 does not have a very complicated structure and could be replaced by a sentence or a list.*

We agree that table 2 clearly could be replaced by a sentence or a list but we prefer a table with the advantage that it is much easier to grasp.

*- The "additional tax of 50% of the current carbon price" on net-negative CO2 emissions (page 29, line 777f) seems to be a very arbitrary modelling choice. Please provide a rationale for this value.*

The rationale for the additional tax of 50% of the current carbon price on net-negative emissions is two-fold: firstly, as soon as total emissions turn net-negative, carbon pricing is not providing revenues anymore but requires net government spending. Secondly, geophysical reasons speak for rather limiting overshooting cumulative emissions budget. The 50% is simply the middle ground between treating net-negative emissions the same as emission reductions or not allowing for net-negative emissions at all, i.e. a tax of 100% which would remove all revenues. The paragraph is changed to "In all regions, an additional tax of 50% of the current carbon price is imposed on net-negative emissions to address two aspects: Firstly, as soon as total emissions turn net-negative, carbon pricing no longer generates revenue but instead requires net government spending. Secondly, geophysical constraints provide grounds for limiting the overshoot of cumulative emissions budget. The 50% assumption is the middle ground between treating net-negative emissions equally to emission

reductions or not allowing for net-negative emissions at all, i.e. a tax of 100% which would preclude any revenues. "

*- The term "internally consistent" may be more intuitive than "self-consistent".*

The term "self-consistent" is changed to "internally consistent" in the revised version of the manuscript.

*- The phrase "investments turn out regrettable" (p13, line 337) and "capital is enlarged" (page 18, line 465) should be revised.*

The first sentence is changed to "As prices change earlier, it turns out that some investments went in the wrong direction (e.g. wrong technology) and adjustments are made in the next period." The second sentence is changed to "Investments increase capital stocks which depreciate according to the depreciation rate...".

*- The sentence "the marginal of the (variable of) taxed activities is impacted by the tax [..]" (page 19, line 499) is not clear.*

We have changed this sentence to "Nevertheless, the marginal value of the variable (but not the parameter) of taxed activities reflects the tax rate which leads to the intended adjustment in the CONOPT solution.".

*- The phrase "these assets are then stranded" (page 23, line 573) should be revised.*

We have changed this sentence to "Capacities are phased out before they reach the end of their technical life-time by the optimization if the value of their outputs is lower than the costs of variable inputs, reflecting a situation of asset stranding. This happens predominantly in 'delayed' scenarios, which begin optimization at a future point in time. "

*- Subsection header 3.4 should be renamed, as this section also includes non-GHG emissions.*

We have changed the header of subsection 3.4 to "Representation of emissions".

*- "roughly a doubling" (page 33, line 850) should be revised.*

We changed the sentence to "... and climate policy in line with 1.5°C results in twice as much deployment compared to …"

*- Section 5 is quite short and the section title "Discussion" is therefore not adequate.*

The title of section 5 is changed to "Conclusions" in the revised version of the manuscript.

*- The programming language(s) should be clearly stated in the section "Code and data availability".*

We added the sentence "The REMIND code is implemented in GAMS while code and data management is done using R." to the section "Code and data availability" in the revised version of the manuscript.

*- Please use the year of the latest update of the model description when citing the IAMC wiki (currently, it does not have a year in the reference)*

We corrected the reference.

**List of main changes made in the manuscript:**

- added section 1.3 on "What is new in REMIND 2.1?"
- provide further explanation for the calibration year 2005 in section 1.4
- added paragraph in section 2 describing the modular structure of REMIND
- reworked section 2.2 for a better explanation of the concept and solution algorithms
- reorganized section 3.2.1 explaining calibration of REMIND
- added further explanation of VRE integration in section 3.2.4
- provide explanation on additional tax in section 3.4.3
- revised section "Code and data availability"
- added Appendix B containing a table of regions
- added Appendix C providing an overview of the most important modules

---

## Author Response (AR2)

Dear topical editor,

Thank you for your helpful comments, shown in blue and italics below. Find our replies directly below each comment.

*1. Section 3.1.2, steady-state vs. equilbrium:*

*I was hoping for a more thorough rewrite of this section - I find it very hard to digest and the overlap with section 2.2 is adding to my confusion. In the last paragraph of Section 3.1.2, you introduce (for the first time) the welfare theorems by Arrow and Debreu, which state that "a competitive market equilibrium can be determined as a Pareto optimum". You then proceed to claim that this is exactly what the Negishi approach does. But in my (maybe wrong?) understanding, a Nash solution (under certain assumptions, which seem to be met by your statement on internalizing an externality) is also the result of a competitive market equilibrium - which (per Section 2.2) differs from the Negishi approach.*

*It could be that trying to explain the solution method before having discussed the underlying economic principles puts the reader onto a very challenging path to follow your exposition. Please reconsider at which point in the manuscript to introduce these concepts, and how to properly introduce the difference beteen equilibrium, general equilibrium and steady state.*

We moved information from section 3.1.2 to section 2.2 for providing economic background when describing the solution methods. Section 3.1.2 is revised and now includes the paragraph "Arrow and Debreu (1954) introduced two welfare theorems with the general equilibrium theory. The so-called Second Welfare Theorem, in particular, states that the market equilibrium can be determined from a Pareto optimum solution. This finding provides the conceptual basis for the Negishi approach, and the market equilibrium is determined from the social planner's solution. Manne and Rutherford (1994) first applied the Negishi approach in an intertemporal setting using a joint maximization algorithm (which is similar to the present algorithm).". Section 3.1.2. is changed to "In economics, the long-term economic growth is called "steady state", meaning the stability of the evolution problem (note: in contrast to physical sciences, "steady state" in the context of macro-economic growth theory means that key characteristics of the system, such as the savings rate, income share of labor, etc., remain constant, while the overall economy still grows). If an economic system is stable, a deviation from the steady state growth path leads to transition processes that close the gap to the steady state (or balanced growth path) asymptotically. During this process the markets are in equilibrium (i.e. prices equal demand and supply) in each time step. This ensures that basic accounting requests are met (i.e. no loss of commodities at the global level). The REMIND model is supposed to analyse transitions to a balanced growth path in response to policies while market equilibrium is ensured at each time (step). The general equilibrium concept on which REMIND is based is mathematically and numerically tractable and the fundamental theoretical framework of a majority of economic models. It aggregates the independent decisions of various economic agents so that production and consumption are consistent, with a balance between supply and demand, which leads to an efficient allocation of goods and services in the economy. Yet, this concept also has some limitations. On the one hand, there are strong assumptions like the perfect information for all agents. On the other hand, uniqueness and robustness of the equilibrium cannot be demonstrated for a very general set

of assumptions (Balasko, 2009). The ability of REMIND to model long-term growth dynamics and ensuing energy demands is hardly contained by limitations of the equilibrium concept. Application of this concept is contained to international trade interactions, while the dynamics of long-term growth is mainly driven by preferences, productivities, technological change, capital accumulation, population growth and endowments (e.g. fossil resources).".

*2. The arbitrary choice of a 50% tax mark-up on net-negative emissions: I understand the rationale for disincentivizing net-negative emissions, but my question concerned the choice of that value. I do not think that "the 50% assumotion is the middle ground" is a very convincing argument.*

The value of 50% is a policy assumption like many other assumptions which are necessary to run scenarios with such a long-horizon model. We changed the sentence to "REMIND assumes the value of 50% to balance the likelihoods that net-negative emissions might be treated equally to emission reductions or not incentivized at all, i.e. a tax of 100% which would preclude any revenues.".

**List of main changes made in the manuscript:**

- reworked section 2.2
- reworked section 3.1.2